# Differentiable Quantum Computing for Large-scale Linear Control

Connor Clayton[*,1,2]    Jiaqi Leng[*,1,3,5]    Gengzhi Yang[*,1,3]

Yi-Ling Qiao[2,4]    Ming C. Lin[2,4]    Xiaodi Wu[†,1,2]

[1]Joint Center for Quantum Information and Computer Science, University of Maryland
[2]Department of Computer Science, University of Maryland
[3]Department of Mathematics, University of Maryland
[4]Center for Machine Learning, University of Maryland
[5]Department of Mathematics and Simons Institute for the Theory of Computing, UC Berkeley
[*]Equal Contribution
[†]xiaodiwu@umd.edu

## Abstract

As industrial models and designs grow increasingly complex, the demand for optimal control of large-scale dynamical systems has significantly increased. However, traditional methods for optimal control incur significant overhead as problem dimensions grow. In this paper, we introduce an end-to-end quantum algorithm for linear-quadratic control with provable speedups. Our algorithm, based on a policy gradient method, incorporates a novel quantum subroutine for solving the matrix Lyapunov equation. Specifically, we build a *quantum-assisted differentiable simulator* for efficient gradient estimation that is more accurate and robust than classical methods relying on stochastic approximation. Compared to the classical approaches, our method achieves a *super-quadratic* speedup. To the best of our knowledge, this is the first end-to-end quantum application to linear control problems with provable quantum advantage.

## 1 Introduction

Over the past few decades, the growing complexity of modern engineering designs has made the control of large-scale dynamical systems a crucial task across various application fields, such as power grid management [56], swarm robotics [16, 18], sensor networks [21], and airline scheduling [57]. These challenges often involve high-dimensional solution spaces with tens of thousands of degrees of freedom, presenting a significant obstacle for traditional optimal control methods.

The emergence of quantum computing has expanded the potential for designing efficient algorithms in numerical optimization and machine learning [1, 36, 62]. By leveraging the principles of quantum mechanics, such as superposition and entanglement, quantum computers excel at efficient data processing, making them promising for accelerating solutions to large-scale computational challenges [31].

Although there has been some progress in quantum algorithms for some specific optimal control problems arising in quantum sciences [37, 39], a viable pathway for accelerating general large-scale optimal control problems remains unclear. A conventional approach to optimal control involves solving the Algebraic Riccati Equation (ARE, see Section 1.1 for details), which is a nonlinear matrix equation. This problem has been less explored in the

field of quantum computing for two reasons. First, most proposed quantum algorithms for algebraic and differential equations focus on linear and vector-valued problems, and extending them to nonlinear matrix equations is highly challenging. Second, while efficient quantum algorithms exist for certain weakly nonlinear problems [41], they are not powerful enough to handle the nonlinearity present in the Algebraic Riccati Equation. A breakthrough in this direction calls for novel ideas in algorithm design.

Inspired by the recent advances in differentiable physics [37, 46, 53] and reinforcement learning [19, 45], we develop an end-to-end quantum algorithm that solves a fundamental optimal control problem called the linear-quadratic regulator (LQR). Given its widely applicable mathematical formulation, LQR has been extensively researched and serves as a standard case study for various computing and learning algorithms [25]; moreover, LQR is of significant practical relevance as many real-world optimal control problems can be formulated to address through linearization techniques. Our quantum algorithm is proven to output an $\varepsilon$-approximate optimal solution in time $\widetilde{\mathcal{O}}\left(n\varepsilon^{-1.5}\right)$[1], where $n$ is the dimension of the state vector and $\varepsilon$ is an error tolerance parameter. The algorithm involves two major components: a quantum differentiable simulator and a quantum-accessible classical data structure. This hybrid quantum-classical framework enables us to employ a *policy gradient* method that exhibits a fast convergence rate for the LQR problem. Since almost all known classical methods for the LQR problem heavily rely on subroutines such as matrix factorization and matrix inversion [6, 32, 34, 35], which require at least $\mathcal{O}(n^3)$ overhead in the problem dimension $n$, our new linear-time quantum algorithm, with *super-quadratic* speedup, offers significant promise for large-scale applications.

**Notation.** We use $\mathbb{R}$ and $\mathbb{C}$ to denote the set of real and complex numbers, respectively. $\mathbb{I}$ denotes an identity operator with an appropriate dimension. For two real vectors $u, v \in \mathbb{R}^n$, the Euclidean inner product $\langle u, v \rangle = u^T v$, and the norm of a vector $u$ is $\|u\| = \sqrt{u^T u}$. Given a symmetric/Hermitian matrix $M$, we denote $\lambda_{\max}(M)$ (or $\lambda_{\min}(M)$) as the maximal/minimal eigenvalue of $M$. The spectral norm of a matrix $M \in \mathbb{R}^{m \times n}$ is denoted by $\|M\| = \sup_{\|v\|=1} \|Mv\|$. The Frobenius norm of a matrix $M \in \mathbb{R}^{m \times n}$ is denoted by $\|M\|_F = \sum_{i,j} |M_{i,j}|^2 = \text{Tr}\left[M^T M\right]$. We say $\xi \sim \mathcal{D}$ if the random variable $\xi \in \mathbb{R}^n$ is distributed according to $\mathcal{D}$.

## 1.1 Problem Formulation

We focus on the infinite-horizon continuous-time linear-quadratic regulator (LQR) problem:

$$\min_{x,u} J = \mathbb{E}\left[\int_0^\infty \left(x^\top(t)Qx(t) + u^\top(t)Ru(t)\right) \mathrm{d}t\right] \tag{1a}$$

$$\text{subject to } \dot{x} = Ax + Bu, \quad x(0) \sim \mathcal{D}, \tag{1b}$$

where $x(t)\colon [0,\infty] \to \mathbb{R}^n$ is the state vector, $u(t)\colon [0,\infty] \to \mathbb{R}^m$ is the control input. $A$ and $B$ are constant matrices of appropriate dimensions; $Q$ and $R$ are positive definite matrices.

**Definition 1.** For a square matrix $M \in \mathbb{R}^{n \times n}$, we say $M$ is *Hurwitz* if every eigenvalue of $M$ has a strictly negative real part.

**Definition 2.** For a controllable pair $(A, B)$, the set of stabilizing feedback gains is given by

$$S_K := \{K \in \mathbb{R}^{m \times n} : A - BK \text{ is Hurwitz}\}. \tag{2}$$

Given a controllable pair $(A, B)$, the optimal controller $u(t)$ of problem (1) can be expressed as a linear function of the state vector $x(t)$, namely

$$u(t) = -K^* x(t), \tag{3}$$

where the matrix $K^*$ is the optimal linear feedback gain. An analytical form of the optimal feedback gain is given by $K^* = R^{-1} B^\top P^*$, where $P^*$ is the unique positive solution to the Algebraic Riccati Equation (ARE),

$$A^\top P + PA + Q - PBR^{-1}B^\top P = 0. \tag{4}$$

---

[1]The $\widetilde{\mathcal{O}}(\cdot)$ notation suppresses the condition number dependence and polylogarithmic factors in $n$ and $\varepsilon$.

For large-scale control problems where the control input is much smaller than the state vector (i.e., $m \ll n$), it is often desired to compute the optimal feedback gain matrix $K^*$ without explicitly solving for $P^*$. To this end, we can rewrite the LQR objective function $J(x, u)$ as a function solely depending on $K$ by leveraging the linearity of the optimal controller $u(t) = -Kx(t)$. With standard algebraic manipulation, it turns out that

$$J(x, u) = f(K) = \begin{cases} \mathrm{Tr}[P(K)\Sigma_0] & K \in \mathcal{S}_K, \\ \infty & \text{otherwise,} \end{cases} \tag{5}$$

where

$$P(K) = \int_0^\infty e^{(A-BK)^\top t} \left( Q + K^\top R K \right) e^{(A-BK)t} \, \mathrm{d}t, \tag{6}$$

and $\Sigma_0 := \mathbb{E}_{\xi \sim \mathcal{D}}[\xi \xi^\top]$. Given this reformulation, the search for the optimal feedback gain $K^*$ reduces to minimizing the unconstrained objective function $f(K)$.

In practice, the matrices $A, B, Q, R$ often possess sparsity structures that can be leveraged by quantum computers. We make the assumptions on the efficient quantum access model.

**Assumption 1** (Sparse-access matrices). We assume $A$, $B$, $Q$, and $R$ are $s$-sparse, i.e., there are at most $s$ non-zero entries in each row/column. For $M \in \{A, B, Q, R\}$, we assume access to an efficient procedure[2] that loads the matrix into quantum data:

$$|i\rangle|k\rangle \mapsto |i\rangle|r_{i,k}\rangle, \quad |\ell\rangle|j\rangle \mapsto |c_{\ell,j}\rangle|j\rangle, \quad |i\rangle|j\rangle|0\rangle \mapsto |i\rangle|j\rangle|M_{ij}\rangle, \tag{7}$$

where $r_{i,k}$ is the index of the $k$-th non-zero entry of the $i$-th row of $M$, $c_{\ell,j}$ is the index of the $\ell$-th non-zero entry of the $j$-th column of $M$, and $M_{ij}$ is a fixed-length binary description of the $(i, j)$-th entry of $M$.

With quantum access to the problem data, we aim to determine the optimal linear feedback gain $K^*$ so that the objective function $f(K)$ is minimized, as summarized in the following problem statement:

**Problem 1.** Assume $(A, B)$ is a controllable pair and $Q$, $R$ are positive-definite. Given quantum access to $A, B, Q, R$ in the sense of Assumption 1, we want to compute an $\varepsilon$-approximate solution $K$ such that $\|K - K^*\|_F \leq \varepsilon$, where $\varepsilon > 0$ is prefixed.

## 1.2 Main Contributions

In this paper, we propose an end-to-end quantum algorithm for solving LQR problems that exhibits the desired quantum advantage in the large-scale setting (i.e., in the parameter regime $m \ll n$). A detailed comparison between ours and various other methods is given in Table 1. Compared with state-of-the-art classical methods, our algorithm achieves a super-quadratic speedup in terms of the state vector dimension $n$. To the best of our knowledge, this is the first end-to-end quantum application to linear control problems with provable speedup.

| Methods | Time/Gate Complexity |
|:---:|:---:|
| Schur method [35, 48] | $\widetilde{\mathcal{O}}(n^3)$ |
| Newton-Kleinman method [32, 47] | $\widetilde{\mathcal{O}}(n^3)$ |
| (Model-based) policy gradient [45] | $\widetilde{\mathcal{O}}(n^3 \cdot \mathrm{poly}(\varepsilon^{-1}))$ |
| **Ours** | $\widetilde{\mathcal{O}}(n\varepsilon^{-1.5})$ (Theorem 7) |

Table 1: Asymptotic cost of different methods for LQR.

Our algorithm is based on a novel policy gradient strategy to find globally optimal solutions for linear-quadratic control problems. A brief overview of the policy gradient method for LQR is available in Section 3.2. In each iteration cycle, our algorithm executes a fast,

---

[2]This input model is sometimes referred to as the *sparse-input oracle* model in the literature. It is a standard assumption in many applications, see [2, 13, 22] for details.

quantum-assisted differentiable simulator to obtain robust gradient estimates, as detailed in Theorem 31. The gradient estimates are then utilized by a classical computer to update the control policy $K$.

As illustrated in Figure 1, with the back-and-forth iterations between the quantum simulator and a classical computer, the control policy $K$ converges to the optimal policy $K^*$ at a linear rate, leading to an end-to-end resolution of the LQR problem.

Our quantum algorithm design can be regarded as a novel realization of the hybrid quantum-classical computing paradigm, where collaboration between classical and quantum computers significantly reduces the burden on the quantum side. Moreover, we provide explicit constructions of the quantum simulator and analyze the convergence rate of the policy gradient based on our end-to-end model. These desirable attributes make our proposed design more practical and relevant in the early fault-tolerant era [29].

Notably, we propose a new quantum algorithm for solving the Lyapunov equation, a fundamental task in optimal control theory [25]. Based on an integral representation of the solution and a rich toolbox of methods for quantum numerical linear algebra, our algorithm can produce a quantum representation of the solution matrix in a cost that

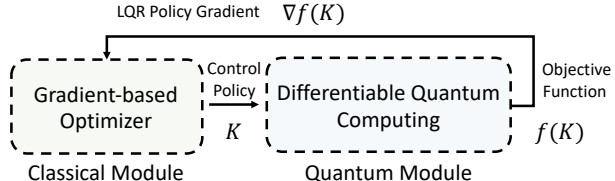

Figure 1: Differentiable quantum computing for linear control.

is polylogarithmic in the matrix dimension $n$ (see Theorem 4), leading to an *exponential speedup* over existing classical methods [51]. Moreover, since the matrix Lyapunov equation is fundamental to many control problems, we foresee that our quantum algorithm may play an important role in finding speedups for other tasks.

The fast quantum algorithm for the Lyapunov equation enables us to develop a quantum gradient estimation subroutine in near-optimal cost, as detailed in Theorem 31. Compared with the conventional gradient estimation techniques based on stochastic approximation (e.g., one- and two-point gradient estimators [45]), our quantum gradient estimation benefits from the explicit exploitation of the analytical form of the gradient. Numerical experiments suggest that our gradient estimation is robust and often leads to faster convergence in practice, as demonstrated in Section 5.3.

## 2  Related Work

We survey related work on model-based and model-free linear-quadratic control, differentiable physics, and quantum reinforcement learning in this section.

**Model-based linear-quadratic control.**  *Model-based* optimal control [17, 50] refers to the scenario where historical measurement data explicitly gives (or estimates) the problem description. In this case, the optimal linear feedback gain $K^*$ can be computed by solving the algebraic Riccati equation (ARE), as detailed in Section 1.1. Commonly used numerical methods for ARE include factorization methods (e.g., Schur method [35, 48]) and iterative methods (e.g., Newton-Kleinman method [20, 32, 47]). These methods require computing matrix factorization or matrix inverse, which in general leads to a $O(n^3)$ run time (assuming $m \leq n$). Some methods for ARE can achieve $\mathcal{O}(n)$ runtime under strong assumptions such as the solution $P^*$ is of low rank [8]. It is also possible to solve LQR by reformulating it as a semidefinite programming (SDP) problem [14]. We do not dive into the SPD approach as it does not demonstrate superior asymptotic scaling compared to other direct methods.

**Model-free linear-quadratic control.**  LQR can be regarded as a continuous-time analog of the discrete Markov Decision Process (MDP) model and many techniques from reinforcement learning (RL) can be introduced to learn the optimal feedback gain (i.e., *control policy*), such as policy gradient [19, 45], natural gradient [25], and policy iteration [10].

These RL-based methods are particularly useful when we have access to the observed costs but the system model can not be directly constructed.

**Differentiable physics and quantum computing.** The differentiable programming paradigm has been applied to many dynamical systems for learning [46] and control [53]. Those gradient-based methods can be used in reinforcement learning [61], inverse problems [27], optimization [5], design [60], etc. People have developed differentiable pipelines for various dynamics including fluids [59], rigid body [49], soft body [26], and other hybrid systems [52]. Recently, [37] have derived a differentiable analog quantum computing pipeline for quantum optimization and control. In this work, we will focus on using differentiable quantum computing to accelerate a widely studied classical problem - linear control synthesis.

**Quantum reinforcement learning.** Recently, quantum-accelerated reinforcement learning has attracted significant attention as it demonstrates the potential for computational speedup [43]. It has been shown that quantum computers can be used to compute policy gradients given coherent access to a Markov Decision tree model [15, 28]. Some works also discuss the quantum policy iteration method for RL, see [12, 58]. It is worth noting that the existing works are usually based on a strong quantum access model and it remains unclear what the cost of constructing such models is in an *end-to-end* sense.

## 3 Preliminaries

### 3.1 Introduction to Quantum Computing

All quantum states of a quantum system form a Hilbert space, which is isomorphic to $\mathbb{C}^N$. We may assume $N = 2^n$ and $n$ is a non-negative integer. An element $|\psi\rangle$ in this Hilbert space is then noted as a $N$-dimensional quantum state, where

$$|\psi\rangle = \begin{bmatrix} v_0 \\ v_1 \\ \vdots \\ v_{N-1} \end{bmatrix}, \tag{8}$$

where $v_i \in \mathbb{C}, i \in \{0, 1, \cdots, N-1\}$. Also, we often use $\langle\psi|$ to denote the conjugate transpose of $|\psi\rangle$. For any $c \neq 0$, $c|\psi\rangle$ and $|\psi\rangle$ refer to the same state, thus without loss of generality, $\||\psi\rangle\| = 1$ always holds. Specifically, a one qubit system corresponds to the aforementioned Hilbert space with $n = 1$.

Given $m$ quantum states $|\psi_1\rangle, |\psi_2\rangle, \cdots, |\psi_m\rangle$ from $m$ quantum systems, then

$$|\psi\rangle = |\psi_1\rangle \otimes |\psi_2\rangle \otimes \cdots \otimes |\psi_m\rangle \tag{9}$$

is a quantum state in the space that consists $m$ subspaces.

The evolution of a quantum state can be described by a unitary operator $U$, meaning

$$U^\dagger U = UU^\dagger = I. \tag{10}$$

We often note these operations as gates on the quantum circuit. One important type of unitary operators are the Pauli operators, namely

$$X = \begin{bmatrix} 0 & 1 \\ 1 & 0 \end{bmatrix}, \quad Y = \begin{bmatrix} 0 & -i \\ i & 0 \end{bmatrix}, \quad Z = \begin{bmatrix} 1 & 0 \\ 0 & -1 \end{bmatrix}. \tag{11}$$

They form a basis of all the linear operators acting on $\mathbb{C}^2$.

For the quantum measurement, given a quantum observable $H$, we can do the measurement of a quantum state $|\psi\rangle$. Specifically, after the measurement on $|\psi\rangle$, the state collapses to $\frac{P_m|\psi\rangle}{\sqrt{p_m}}$, and we get an outcome $\lambda_m$ with probability $p_m = \langle\psi|P_m|\psi\rangle$, where

$$H = \sum_m \lambda_m P_m \tag{12}$$

is the spectral decomposition of $H$.

## 3.2 Policy Gradient for LQR

For all stabilizing feedback gains $K \in S_K$, the gradient of the objective function $f(K)$ as defined in (5) has the following closed-form expression [38, 40]:

$$\nabla f(K) = 2(RK - B^\top P(K))X(K), \tag{13}$$

where $P(K)$ is given in (6), and $X(K)$ is determined by

$$X(K) = \int_0^\infty e^{(A-BK)t} \Sigma_0 e^{(A-BK)^\top t} \, \mathrm{d}t. \tag{14}$$

The (direct) policy gradient method for LQR minimizes the objective $f(K)$ via the vanilla gradient update rule $K \leftarrow K - s\nabla f(K)$, where $s > 0$ is a fixed step size. Given sufficiently small $s$, it has been shown that the policy gradient method converges at a linear rate [45, Theorem 2]. In practice, however, the policy gradient is often estimated through stochastic approximation, such as one- and two-point estimation [45]. While these zeroth-order gradient estimation methods are less demanding in terms of computational cost, they tend to be sensitive to random perturbations and slow to converge.

In this paper, we propose a fast quantum algorithm that outputs a robust estimate of the gradient in $\widetilde{\mathcal{O}}(n)$ time (assuming $m \ll n$, see Theorem 31). Leveraging the quantum gradient estimation subroutine, we recover the linear convergence rate using robust gradient descent, as detailed in Proposition 34.

## 3.3 Quantum Data Structure

To perform policy gradient in the training process, the linear feedback gain $K$ is stored in a *quantum-accessible data structure* as proposed in [30]. This data structure allows intermediate updates on $K$ and efficient quantum queries to $K$ as a block-encoded matrix. This data structure is a purely classical representation of $K$, and quantum access to this data structure (e.g., through qRAM [23]) is required to build the block-encoding of $K$. In the literature, this data structure is also known as *classical-write, quantum-read* qRAM [7, 54].

**Definition 3** (Block-encoding). Suppose that $M$ is an $p$-qubit operator, $\alpha, \varepsilon \in \mathbb{R}^+$ and $r \in \mathbb{N}$, then we say that the $(p + r)$-qubit unitary $U_M$ is an $(\alpha, r, \varepsilon)$-block-encoding of $M$, if

$$\|M - \alpha(\langle 0|^r \otimes \mathbb{I})U_M(|0\rangle^r \otimes \mathbb{I})\| \leq \varepsilon. \tag{15}$$

In this paper, the growth of ancilla qubits (space complexity) is dominated by the number of elementary gates (gate complexity). Therefore, when referring to a specific block-encoding, we often omit the number of ancilla qubits (i.e., the parameter $r$) for simplicity.

**Lemma 1.** *Let $K \in \mathbb{R}^{m \times n}$. There exists a data structure to store $K$ with the following properties: (1) the size of the data structure is $\mathcal{O}\left(mn \log^2(mn)\right)$, (2) the time to store a new entry $(i, j, \hat{K}_{i,j})$ is $\mathcal{O}\left(\log^2(mn)\right)$, and (3) for any $\varepsilon > 0$, a quantum algorithm can implement a $\left(\|K\|_F, \lceil \log_2 n \rceil + 2, \varepsilon\right)$-block-encoding of $K$ in time $\mathcal{O}\left(\mathrm{poly} \log(n, 1/\varepsilon)\right)$. There also exists an analogous data structure for $\hat{K}^\top$.*

*Proof.* We use the data structure as described in [30, Theorem 5.1]. To construct the block-encoding, we utilize [22, Lemma 50]. $\qquad\square$

## 3.4 Quantum Simulation of Linear Dynamics

Quantum computers can simulate certain linear ordinary differential equations (ODEs) exponentially faster than classical computers [3, 4, 9, 22, 33]. In this paper, we present a quantum simulation subroutine based on quantum linear system solvers (QLSS) [9, 33], as described below. While this approach may not be optimal in terms of state preparation cost compared to quantum singular value transformation [22] or linear combination of Hamiltonian simulation [3, 4], it allows us to incorporate the Hurwitz stability of the system.

**Theorem 2** (Informal version of Theorem 17). *Suppose that $\mathcal{A} \in \mathbb{R}^{n \times n}$ is a Hurwitz matrix, and $O_{\mathcal{A}}$ is an $(\alpha, 0)$-block-encoding of $\mathcal{A}$. For an arbitrary $t > 0$, we can implement a $(\zeta t, \varepsilon)$-block-encoding of $e^{\mathcal{A}t}$ using $\widetilde{\mathcal{O}}(\alpha \rho t \cdot \mathrm{poly} \log(1/\varepsilon))$ queries to $O_A$, and $\widetilde{\mathcal{O}}(\alpha \rho t \cdot \mathrm{poly} \log(1/\varepsilon))$ queries to additional gates. Here, the normalization factor $\zeta = \mathcal{O}(\alpha \rho)$, and the constant $\rho$ is solely determined by $\mathcal{A}$.*

More details and the proof of Theorem 2 can be found in Appendix C. Note that the dependence on $t$ in the above result can be further improved using a standard padding technique, but for simplicity, we do not discuss this minor improvement, as it does not affect our main end-to-end result. We also notice a technique called quantum eigenvalue transformation (QEVT), recently proposed by Low and Su [42]. While this method cannot be directly applied to Hurwitz-stable systems, it may be enhanced to provide a simulation algorithm with a similar cost, as discussed in Appendix D.

## 4 Quantum Algorithm for the Lyapunov Equation

The (continuous-time) Lyapunov equation is a linear matrix equation of the following form,

$$\mathcal{A}X + X\mathcal{A}^{\top} + \Omega = 0. \tag{16}$$

Since this equation is linear in terms of the matrix $X$, it is possible to derive a vectorization form of (16) and solve it using a quantum linear system algorithm [11, 24]. In this paper, we propose a new quantum algorithm for solving the Lyapunov equation based on an integral representation formula. Our algorithm directly prepares a block-encoded solution matrix $X^*$. Compared to the previous approach, our method leads to an exponentially faster quantum objective function evaluation algorithm (see Theorem 5) and a new quantum gradient estimation subroutine (see Theorem 6).

### 4.1 Representation

Given a positive-definite $Q$, there exists a unique positive-definite $X^*$ satisfying (16) if and only if $\mathcal{A}$ is Hurwitz. The unique positive solution is given by

$$X^* = \int_0^{\infty} e^{\mathcal{A}t} \Omega e^{\mathcal{A}^{\top} t} \, \mathrm{d}t. \tag{17}$$

The integral formula (17) suggests that the solution to the Lyapunov equation can be computed using a numerical integration technique. For a finite $\tau > 0$, we define

$$X_{\tau} := \int_0^{\tau} e^{\mathcal{A}t} \Omega e^{\mathcal{A}^{\top} t} \, \mathrm{d}t. \tag{18}$$

For any arbitrary $\varepsilon > 0$, we find that a $\tau = \widetilde{\mathcal{O}}(\log(1/\varepsilon))$ is sufficient to ensure an $\varepsilon$-approximate solution. We denote $\kappa := \|X^*\|/\lambda_{\min}(\Omega)$.

**Lemma 3** (Numerical integration). *For any $\varepsilon > 0$, we have $\|X^* - X_{\tau}\| \leq \varepsilon$, provided that*

$$\tau = \kappa \log\left(\frac{\|\Omega\| \|X^*\| \kappa}{\varepsilon \lambda_{\min}(X^*)}\right). \tag{19}$$

*Proof.* Note that $\|X^* - X_{\tau}\| = \left\|\int_{\tau}^{\infty} e^{\mathcal{A}t} \Omega e^{\mathcal{A}^{\top} t} \, \mathrm{d}t\right\| \leq \int_{\tau}^{\infty} \|\Omega\| \|e^{\mathcal{A}t}\| \|e^{\mathcal{A}^{\top} t}\| \, \mathrm{d}t$, where $\|e^{\mathcal{A}t}\|$ (or $\|e^{\mathcal{A}^{\top} t}\|$) is upper bounded by $\frac{\|X^*\|}{\lambda_{\min}(X^*)} e^{-t/\kappa}$ (see [44, Lemma 12]). It follows that $\|X^* - X_{\tau}\| \leq \frac{\|\Omega\| \|X^*\| \kappa}{\lambda_{\min}(X^*)} e^{-\tau/\kappa}$. Therefore, an integration time as given in (19) guarantees that $\|X^* - X_{\tau}\| \leq \varepsilon$. $\qquad \square$

## 4.2 Algorithm Complexity Analysis

The matrix $X_\tau$ can be approximated by a trapezoidal rule with $(K+1)$ quadrature node points, namely,

$$X_\tau \approx \sum_{k=0}^{K} w_k e^{\mathcal{A} t_k} \Omega e^{\mathcal{A}^\top t_k} = \sum_{k=0}^{K} w_k \mathscr{F}(t_k), \tag{20}$$

where $w_k = \frac{(2 - 1_{k=0,K}) \tau}{2K}$, $t_k = \frac{k\tau}{K}$, and $\mathscr{F}(t) := e^{\mathcal{A} t} \Omega e^{\mathcal{A}^\top t}$. The summation in (20) can be computed on a quantum computer by performing linear combinations of block-encoded matrices, which we will explain shortly.

**Definition 4** (Select oracle). Let $\mathcal{A} \in \mathbb{R}^{n \times n}$ be a Hurwitz matrix. Given an integer $K > 0$ and two positive scalars $\tau, \varepsilon > 0$, we define the following unitary (named as the *select oracle*):

$$\mathrm{select}(\mathcal{A}, \varepsilon) := \sum_{k=0}^{K} |k\rangle\langle k| \otimes U_k, \tag{21}$$

where for each $k = 0, \dots, K$, $U_k$ is a $(\zeta t_k, \varepsilon)$-block-encoding of $e^{\mathcal{A} t_k}$ with $t_k = k\tau/K$. Here, $\zeta$ denotes some parameter that only depends on $\mathcal{A}$.

Now, we consider two select oracles:

$$\mathrm{select}(A, \varepsilon) := \sum_{k=0}^{K} |k\rangle\langle k| \otimes U_k, \quad \mathrm{select}(A^\top, \varepsilon) := \sum_{k=0}^{K} |k\rangle\langle k| \otimes V_k, \tag{22}$$

where for each $k = 0, \dots, K$, $U_k$ (or $V_k$) denotes a block-encoding of $e^{\mathcal{A} t_k}$ (or $e^{\mathcal{A}^\top t_k}$) with normalization factor $\zeta t_k$. Let $O_\Omega$ be a $(\eta, 0)$-block-encoding of $\Omega$, and we find that

$$\mathrm{select}(A, \varepsilon)(I \otimes O_\Omega)\mathrm{select}(A^\top, \varepsilon) = \sum_{k=0}^{K} |k\rangle\langle k| \otimes W_k, \tag{23}$$

where $W_k := U_k O_\Omega V_k$ is a $(\eta\zeta^2 t_k^2, 2\zeta\eta\varepsilon)$-block-encoding of the matrix $\mathscr{F}(t_k)$. Denoting $\lambda_k := w_k k^2$, then it is clear that $\sum_{k=0}^{K} \lambda_k W_k$ is a block-encoding of $X_\tau$. Thus we can implement a block-encoded $X_\tau$ on a quantum computer using a technique known as linear combination of unitaries (LCU) [22, Lemma 52]. The rigorous complexity of this quantum algorithm is given in the following theorem, for which a complete proof is provided in Appendix E.

**Theorem 4.** *Suppose that $\mathcal{A} \in \mathbb{R}^{n \times n}$ is Hurwitz and $\Omega \in \mathbb{R}^{n \times n}$ is positive-definite. Let $O_\mathcal{A}$ be an $(\alpha, 0)$-block-encoding of $\mathcal{A}$ and $O_\Omega$ be an $(\eta, 0)$-block-encoding of $\Omega$. Then, we can implement a $(\gamma, \varepsilon)$-block-encoding of $X^*$, the unique solution to the Lyapunov equation (16), using a single query to $O_\Omega$, $\widetilde{\mathcal{O}}\left(\alpha^2\sqrt{\frac{\eta}{\varepsilon}}\right)$ queries to controlled $O_\mathcal{A}$ and its inverse, and $\widetilde{\mathcal{O}}\left(\alpha^2\sqrt{\frac{\eta}{\varepsilon}}\right)$ queries to other additional elementary gates. Here $\gamma = \widetilde{\mathcal{O}}(\alpha^2\eta)$.*

## 4.3 Objective Function Evaluation

As a direct consequence of Theorem 4, we can evaluate the objective function value $f(K)$ for a given $K \in \mathcal{S}_K$ in a cost that is logarithmic in the dimension parameter $n$. This result demonstrates an exponential quantum advantage for the objective function evaluation task, as any known classical algorithm for this task requires at least matrix multiplication time.

**Theorem 5.** *Assume that we have efficient procedures (as described in Assumption 1) to access the problem data $A, B, Q, R$ in $\mathcal{O}(\mathrm{poly}\log(n))$ time. Let $K \in \mathcal{S}_K$ be a stabilizing policy stored in a quantum-accessible data structure. We can estimate the objective function $f(K)$ up to multiplicative error $\theta$ in cost $\widetilde{\mathcal{O}}\left(\frac{1}{\theta^2}\right)$.*

*Proof.* Given a $K \in \mathcal{S}_K$, we can evaluate the objective function $f(K)$ via the formula $f(K) = \mathrm{Tr}[P(K)\Sigma_0]$. Here, without loss of generality, we assume $\Sigma_0 = \mathbb{I}$. $P(K)$ has a closed-form representation as in (6), which corresponds to the unique positive solution to the Lyapunov equation

$$(A - BK)^\top P + P(A - BK) + (Q + K^\top RK) = 0. \tag{24}$$

We denote $\mathcal{A} = A - BK$, and $\Omega = Q + K^\top RK$. By Lemma 11 and Remark 1, we can block-encode $\mathcal{A}$ with normalization factor $s(\|K\|_F + 1)$ and $\Omega$ with normalization factor $s(\|K\|_F^2 + 1)$, both in cost $\mathcal{O}(\text{poly}\log(n, s))$. Suppose that $f(K) = a$, it follows from Lemma 8 that $\|K\|_F \leq \mathcal{O}(a)$. Therefore, by Theorem 4, we can implement a $(\gamma, \varepsilon)$-block-encoding of $P(K)$ in cost $\tilde{\mathcal{O}}\left(a^3\rho\sqrt{\frac{\kappa^5}{\varepsilon}}\right)$, where $\gamma \leq \widetilde{\mathcal{O}}(a^4\rho^2\kappa^3)$. Note that $P(K)$ is a Hermitian matrix, and $\lambda_{\min}(P(K)) \geq \lambda_{\min}(Q)$, by invoking Theorem 25, we can estimate $f(K) = \text{Tr}[P(K)]$ up to a multiplicative error $\theta$ in cost $\widetilde{\mathcal{O}}\left(\frac{a^3\rho}{\theta}\sqrt{\frac{\gamma^3\kappa^5}{\varepsilon}}\right) \leq \widetilde{\mathcal{O}}\left(\frac{a^{13}\rho^6\kappa^{10}}{\theta^2}\right)$. Here, the error parameter must be chosen so that $\varepsilon \leq \widetilde{\mathcal{O}}(\theta^2/\gamma^2)$. $\qquad\square$

# 5 Quantum Policy Gradient for Large-Scale Control

## 5.1 Quantum Gradient Estimation

**Definition 5.** Given any $K \in \mathcal{S}_K$, we call $G \in \mathbb{R}^{m \times n}$ a *$\theta$-robust estimate* of $\nabla f(K)$ if it approximates the gradient $\nabla f(K)$ up to a multiplicative error $\theta$, i.e.,

$$\|G - \nabla f(K)\|_F \leq \theta\|\nabla f(K)\|_F. \tag{25}$$

Here, we utilize the close-form expression of the policy gradient $\nabla f(K)$, as shown in (13), to construct a quantum algorithm for gradient estimation. The complete theorem statement and the proof are in Appendix G.

**Theorem 6** (Informal version of Theorem 31)**.** *Assume we have efficient procedures (as described in Assumption 1) to access $A, B, Q, R$ in $\mathcal{O}(\text{poly}\log(n))$ time. Let $K \in \mathcal{S}_K$ be a stabilizing policy stored in a quantum-accessible data structure. Provided that $\|K - K^*\| > \varepsilon$ and $m \ll n$, we can compute a $\theta$-robust estimate of $\nabla f(K)$ in cost $\widetilde{\mathcal{O}}\left(\frac{n}{\theta^{1.5}\varepsilon^{1.5}}\right)$.*

## 5.2 Quantum Policy Gradient

Our main quantum algorithm for LQR is summarized in Algorithm 1.

---
**Algorithm 1** Quantum policy gradient

---
**Inputs:** $A, B, Q, R$ (problem data), $K_0 \in \mathcal{S}_K$ (initial guess), $\sigma > 0$ (step size/learning rate), $\theta$ (robustness parameter), $N$ (number of iterations)
**Output:** an approximate solution $K_N$

---
   **for** $k \in \{0, 1, \ldots, N-1\}$ **do**
      Compute a $\theta$-robust estimate of $\nabla f(K_k)$, denoted as $G_k$, using Theorem 6.
      Update the quantum-accessible data structure using the rule: $K_{k+1} = K_k - \sigma G_k$.
   **end for**

---

We can prove that the iterates in Algorithm 1 converges to the optimal control policy $K^*$ at a linear rate (Proposition 34). It follows that our algorithm can find an $\varepsilon$-approximate optimal policy with an end-to-end cost $\widetilde{\mathcal{O}}\left(\frac{n}{\varepsilon^{1.5}}\right)$.

**Theorem 7** (Informal version of Theorem 35)**.** *Assume that we have efficient procedures (as described in Assumption 1) to access the problem data $A, B, Q, R$ in $\mathcal{O}(\text{poly}\log(n))$ time. Let $K_0 \in \mathcal{S}_K$ be a stabilizing policy and assume that $m \ll n$. Then, Algorithm 1 outputs an $\varepsilon$-approximate solution to Problem 1 in cost $\widetilde{\mathcal{O}}\left(\frac{n}{\varepsilon^{1.5}}\right)$.*

## 5.3 Numerical Experiments

**Correctness.** We conduct a numerical experiment to showcase the correctness of our quantum policy gradient algorithm. Following a similar setup as in [45], a mass-spring-damper system with $g = 4$ masses is used for constructing our LQR problem. The state

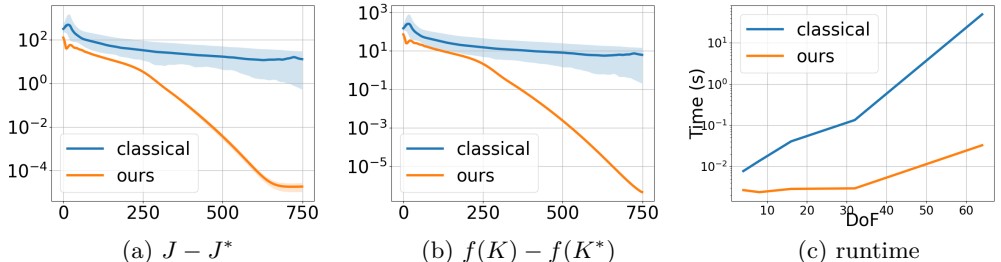

(a) $J - J^*$

(b) $f(K) - f(K^*)$

(c) runtime

Figure 2: **Numerical Results on Convergence.** Following the mass-spring-damper setup in [45], our policy gradient descent algorithm converges much faster than [45].

$x = [p^\top, v^\top]^\top \in \mathbb{R}^{2g}$ contains positions and velocities, with dynamic and input matrices,

$$A = \begin{bmatrix} 0 & I \\ -T & -T \end{bmatrix}, B = \begin{bmatrix} 0 \\ I \end{bmatrix}, Q = I + 100e_1 e_1^\top, R = I + 4e_2 e_2^\top, \tag{26}$$

where $0, I$ are $g \times g$ zero and identity matrices, $e_i$ is the $i$th unit vector, and matrix $T$ has 2 on the main diagonal and -1 on the first super- and sub-diagonal. In Figure 2, we run our method against the classical model-free gradient-based method [45]. It shows that our model-based policy gradient converges much faster to the ground truth ARE solution $K^*$. In the benchmark example [45], ours converges within 750 iterations, while the classical method takes $2 \times 10^4$ iterations (*orders of magnitude* longer). In Figure 2 (c), we increase the system size by scaling $g$ from 4 to 64. Both methods run on a classical simulator with Intel i9-10980XE CPU. Our method runs much faster than [45] by **nearly 3 orders of magnitude**. The code for both methods can be seen at `https://github.com/YilingQiao/diff_lqr`. Additional numerical results can be found in Appendix I.

## 6  Conclusion and Future Work

In this paper, we propose the first quantum algorithm for solving linear-quadratic control problems that achieves end-to-end quantum speedups. Our quantum algorithm utilizes an exponentially faster quantum linear dynamics simulator combined with a policy gradient method. Compared to classical approaches relying on matrix factorization and iterations, our method achieves super-quadratic speedup in the large-scale regime (i.e., $m \ll n$). Moreover, the hybrid quantum-classical algorithm design makes our algorithm a promising candidate for practical quantum advantage in the near horizon. We also provide numerical evidence to demonstrate the robustness and favorable convergence behavior of our method.

**Limitations and Future Work.** Accelerating optimal control and reinforcement learning using quantum computers remains an emerging research topic. Our work has focused on the theoretical aspects of quantum advantage for LQR, a classic optimal control problem of fundamental importance in both theory and practice. However, for special cases [17], we have no guarantee that our quantum algorithm still applies, since Lemma 33 may not hold. In the future, we aim to explore both the practical utility of quantum computing for such tasks and its potential for handling more complex optimal control scenarios, such as non-quadratic and nonlinear problems.

## Acknowledgment

We thank Kaiqing Zhang for the helpful discussions and the anonymous reviewers for their helpful feedback. This work was supported in part by the U.S. Department of Energy Office of Science, National Quantum Information Science Research Centers, Quantum Systems Accelerator, Air Force Office of Scientific Research (Award#FA9550-21-1-0209), the U.S. National Science Foundation Career Grant (CCF-1942837), a Sloan research fellowship, Meta Research PhD Fellowship, National Science Foundation Graduate Research Fellowship (Grant No. DGE 2236417), the Simons Quantum Postdoctoral Fellowship, a Simons Investigator award (Grant No. 825053), and Dr. Barry Mersky & Captial One E-Nnovate Endowed Professorships.

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

# Appendices

## A  LQR theory

**Definition 6.** Given a positive number $a > 0$, we define the sublevel set $S_K(a) := \{K \in \mathbb{R}^{m \times n} \colon f(K) \leq a\}$. We have the following useful bound on $K$.

**Lemma 8.** *Over the sublevel set $S_K(a)$ of the LQR objective function $f(K)$, we have*

$$\mathrm{Tr}[X(K)] \leq a/\lambda_{\min}(Q), \tag{27a}$$

$$\nu/a \leq \lambda_{\min}(X(K)), \tag{27b}$$

$$\|K\|_F \leq a/\sqrt{\nu\lambda_{\min}(R)}, \tag{27c}$$

*where the constant*

$$\nu = \frac{1}{4}\left(\frac{\|A\|_2}{\sqrt{\lambda_{\min}(Q)}} + \frac{\|B\|_2}{\sqrt{\lambda_{\min}(R)}}\right)^{-2}. \tag{28}$$

*Proof.* See [45, Lemma 16]. $\qquad\square$

**Lemma 9.** *For any $K \in \mathcal{S}_K(a)$, we have*

$$\|K - K^*\|_F^2 \leq \frac{a}{\nu\lambda_{\min}(R)}\left(f(K) - f(K^*)\right), \tag{29}$$

*where $\nu$ is the same as in (28).*

*Proof.* By [45, Lemma 2], we have

$$f(K) - f(K^*) = \mathrm{Tr}\left[(K - K^*)^\top R(K - K^*)X(K)\right] \geq \lambda_{\min}(R)\lambda_{\min}(X(K))\|K - K^*\|_F^2.$$

Combining the above result with Lemma 8, we end up with (29). $\qquad\square$

**Lemma 10** (PL condition). *Fix $a > 0$. For any $K \in \mathcal{S}_K(a)$, we have*

$$\|\nabla f(K)\|_F^2 \geq 2\mu_f(f(K) - f(K^*)), \tag{30}$$

*where $\mu_f > 0$ is a constant that only depends on the problem data and $a$.*

*Proof.* See [45, Remark 2]. $\qquad\square$

## B  Implementation of block-encoded matrices

**Lemma 11.** *Assume that we have efficient procedures (as described in Assumption 1) to access the problem data $A, B, Q, R$ in $\mathcal{O}(\mathrm{poly}\log(n))$ time. For a fixed $a > 0$, suppose that $K \in \mathcal{S}_K$ is a stabilizing policy stored in a quantum-accessible data structure. Then, we can implement*

1. *a $(s(\|K\|_F + 1), \varepsilon)$-block-encoding of $A - BK$ in cost $\mathcal{O}(\mathrm{poly}\log(n, s/\varepsilon))$,*

2. *a $(s(\|K\|_F^2 + 1), \varepsilon)$-block-encoding of $Q + K^\top RK$ in cost $\mathcal{O}(\mathrm{poly}\log(n, s/\varepsilon))$.*

*Proof.* By [22, Lemma 48], we can implement a $(s, \varepsilon)$-block-encoding of $A$ (or $B, Q, R$) in cost $\mathcal{O}(\mathrm{poly}\log(n, s/\varepsilon))$. Also, due to [22, Lemma 50], we can implement a $(\|K\|_F, \varepsilon)$-block-encoding of $K$ in cost $\mathcal{O}(\mathrm{poly}\log(n, 1/\varepsilon))$. Therefore, by [22, Lemma 52, 53], a $(s(\|K\|_F + 1), \varepsilon)$-block-encoding of $A - BK$ can be implement in cost $\mathcal{O}(\mathrm{poly}\log(n, s/\varepsilon))$. Similarly, we can implement a $(s(\|K\|_F^2 + 1), \varepsilon)$-block-encoding of $Q + K^\top RK$ in cost $\mathcal{O}(\mathrm{poly}\log(n, s/\varepsilon))$. $\qquad\square$

**Remark 1.** We observe that the block-encodings of $A - BK$ and $Q + K^\top RK$ can be implemented with high precisions, i.e., in cost $\mathcal{O}(\mathrm{poly}\log(1/\varepsilon))$. To simplify our technical arguments, we assume these block-encodings can be implemented with no error, i.e., we can implement a $(s(\|K\|_F + 1), 0)$-block-encoding of $A - BK$ (or a $(s(\|K\|_F^2 + 1), 0)$-block-encoding of $Q + K^\top RK$ in cost $\mathcal{O}(\mathrm{poly}\log(n, s))$.

## C  Matrix exponential based on quantum linear system solver

### C.1  Block-encoding for matrix inverse

**Lemma 12** (Modified from [42], Lemma 11)**.** *Let $C$ be a matrix such that $C/\alpha_C$ is block-encoded by $O_C$ with some normalization factor $\alpha_C$. Then we can implement a $(\mathcal{O}(\alpha_{C^{-1}}), \varepsilon)$-block-encoding of $C^{-1}$ using*

$$\mathcal{O}(\kappa_C \log(1/\varepsilon)) \tag{31}$$

*queries to $O_C$. Here $\kappa_C$ is the condition number of $C$.*

Suppose we have an $(\alpha, \varepsilon)$-block-encoding $U_M$ that block-encodes $M$. Denote

$$\alpha \langle 0 | U_M | 0 \rangle - M = \Lambda, \tag{32}$$

we have $\|\Lambda\| \leq \varepsilon$. Then we can see $U_M$ as an $(\alpha, 0)$-block-encoding of $M + \Lambda/\alpha$. So with the lemma above, we have the theorem below:

**Theorem 13.** *Suppose we have an $(\alpha, \varepsilon_1)$-block-encoding $U_M$ that block-encodes $M$, then we can implement a $(\mathcal{O}(\alpha_{M^{-1}}), \varepsilon_2 + \frac{\alpha_{M^{-1}}^2}{\alpha - \varepsilon_1 \alpha_{M^{-1}}} \varepsilon_1)$-block-encoding for $M^{-1}$ using*

$$\mathcal{O}(\kappa_M \log(1/\varepsilon_2)) \tag{33}$$

*queries to $U_M$.*

*Proof.* From the lemma above we know we can implement a $(\tilde{\alpha}_{M^{-1}}, \varepsilon_2)$-block-encoding as $\tilde{U}_{M^{-1}}$ for $(M + \Lambda/\alpha)$, where $\tilde{\alpha}_{M^{-1}} = \mathcal{O}(\alpha_{M^{-1}})$. To analyze the error,

$$\|\tilde{\alpha}_{M^{-1}} \tilde{U}_{M^{-1}} - M^{-1}\| \leq \varepsilon_2 + \|(M + \Lambda/\alpha)^{-1} - M^{-1}\| = \varepsilon_2 + \|(I + M^{-1}\Lambda/\alpha)^{-1} M^{-1} - M^{-1}\|$$

$$= \varepsilon_2 + \|(I + M^{-1}\Lambda/\alpha)^{-1} - I\| \|M^{-1}\| = \varepsilon_2 + \left\|\sum_{n=1}^{\infty} ((-M^{-1}\Lambda)/\alpha)^n\right\| \|M^{-1}\|$$

$$\leq \varepsilon_2 + \sum_{n=1}^{\infty} \left\|((-M^{-1}\Lambda)/\alpha)^n\right\| \|M^{-1}\| \leq \varepsilon_2 + \frac{\alpha_{M^{-1}}^2}{\alpha - \varepsilon_1 \alpha_{M^{-1}}} \varepsilon_1.$$

$$\tag{34}$$

$\square$

### C.2  Introduction to quantum linear system solver

The first quantum algorithm for solving linear differential equations was proposed by [9]. Since this work was done, several refinements have been made. Among these, [33] significantly loosen the requirement of performing the algorithm. We notice that this whole algorithm can be written in the form of block-encoding. Basically, for a linear differential equation

$$\frac{\mathrm{d}u}{\mathrm{d}t} = Au, \tag{35}$$

upon appropriate discretization method and the method of line, one may construct a big matrix $\mathbf{A}$ s.t.

$$\mathbf{A} \begin{bmatrix} u(0) \\ u(h) \\ u(2h) \\ \vdots \\ u(T) \end{bmatrix} = \begin{bmatrix} u(0) \\ 0 \\ 0 \\ \vdots \\ 0 \end{bmatrix}. \tag{36}$$

Then it is easy to see that the matrix inverse $\mathbf{A}^{-1}$ satisfies

$$(\langle k | \otimes I) \mathbf{A}^{-1} (|0\rangle \otimes u(0)) = u(kh) = e^{khA} u(0). \tag{37}$$

Because $u(0)$ is arbitrarily chosen, we conclude

$$(\langle k | \otimes I) \mathbf{A}^{-1} (|0\rangle \otimes I) = e^{khA}. \tag{38}$$

Suppose $\mathbf{A}^{-1}$ is block-encoded in some $U_{\mathbf{A}^{-1}}$, we have

$$(\langle k | \otimes I)(((\langle 0^a | \otimes I) U_{\mathbf{A}^{-1}} (|0^a\rangle \otimes I))(|0\rangle \otimes I) = (\langle k 0^a | \otimes I) U_{\mathbf{A}^{-1}} (|0^{a+\ell}\rangle \otimes I). \tag{39}$$

Here $\ell$ satisfy $2^\ell = T/h$, where $T$ is the simulation time and $h$ is the step size. Then it is obvious that $U_{\mathbf{A}^{-1}}$ is also a block-encoding of $e^{khA}$.

### C.3 Matrix exponential construction

**Lemma 14** (Modified from [33]). *If we have a block-encoding $U_L$ that block-encodes $L$ defined as*

$$L = I - N,$$

$$N = \sum_{i=0}^{m} |i+1\rangle\langle i| \otimes M_2(I - M_1)^{-1} + \sum_{i=m+1}^{2m} |i+1\rangle\langle i| \otimes I,$$

$$M_1 = \sum_{j=0}^{k-1} |j+1\rangle\langle j| \otimes \frac{Ah}{j+1}, \tag{40}$$

$$M_2 = \sum_{j=0}^{k} |0\rangle\langle j| \otimes I,$$

*then the $(\alpha_{L^{-1}}, \varepsilon)$-block-encoding $U_{L^{-1}}$ that block-encodes $L^{-1}$ satisfies*

$$\left\| \alpha_{L^{-1}}(\langle r0^a| \otimes I)U_{L^{-1}}(|0^{a+s}\rangle \otimes I) - e^{TA} \right\| \le \epsilon. \tag{41}$$

*for any $r \ge m+1$. Let $2^{s-1} = m$, thus*

$$\alpha_{L^{-1}}(\langle 0^{a+s}| \otimes I)\left((X \otimes I_{s-1} \otimes I_a \otimes I)U_{L^{-1}}\right)(|0^{a+s}\rangle \otimes I) \approx e^{TA}, \tag{42}$$

*which means that $((X \otimes I_{s-1} \otimes I_a \otimes I)U_{L^{-1}})$ is a block-encoding of $e^{TA}$ with its normalization factor as $\alpha_{L^{-1}}$ and error being at most $\varepsilon$.*

*Proof.* The proof of the correctness of $L$ can be found in [33]. ∎

Now we turn to construct the block-encoding for $L$.

**Lemma 15.** *Assume we can query the $(\alpha_A, 0)$-block-encoding $U_A$ that encodes $A$ and $h\alpha_A = \mathcal{O}(1)$, then we can construct a $(\mathcal{O}(k^{1.5}), k^{1/2}\epsilon)$-block-encoding for $L$ defined in (40) using $\mathcal{O}(k\log(1/\epsilon))$ queries of $U_A$ and same queries to additional elementary gates.*

*Proof.* First we need to block-encode $M_1$. We will use the ADD operator defined as

$$\text{ADD} := \sum_j |(j+1) \mod k\rangle\langle j|, \tag{43}$$

and the controlled-rotation $U_R$ as

$$U_R|j+1\rangle|0\rangle = |j+1\rangle\left(\frac{1}{j+1}|0\rangle + \sqrt{1 - \frac{1}{(j+1)^2}}|1\rangle\right). \tag{44}$$

Notice that

$$(U_R \otimes I)(\text{ADD} \otimes I \otimes I)\left(\sum_{j=0}^{k-1} |j\rangle\langle j| \otimes I \otimes U_A\right)$$
$$= \sum_{j=0}^{k-1} U_R\left(|j+1 \mod k\rangle\langle j| \otimes I\right) \otimes U_A. \tag{45}$$

Post-select on the second register and the ancilla qubits of $U_A$, we have

$$(I \otimes \langle 0| \otimes \langle 0|)\left(\sum_{j=0}^{k-1} U_R\left(|j+1 \mod k\rangle\langle j| \otimes I\right) \otimes U_A\right)(I \otimes |0\rangle \otimes |0\rangle)$$
$$= \sum_{j=0}^{k-1} |(j+1) \mod k\rangle\langle j| \otimes \frac{A/\alpha_A}{j+1}. \tag{46}$$

If we apply the operator on a state, namely

$$\left( \sum_{j=0}^{k-1} U_R \big( |(j+1) \mod k \rangle \langle j| \otimes I \big) \otimes U_A \right) |t\rangle |0\rangle |\psi\rangle = \frac{1}{\alpha_A(t+1)} |(t+1) \mod k \rangle |0\rangle A|\psi\rangle + |\perp\rangle.$$

(47)

So we can apply one more multi-controlled-$X$ gate to flip the flag register for the control register being $|0\rangle$, then the whole thing becomes a $(\alpha_A h, 0)$-block-encoding for $M_1$, using just one query to $U_A$. We can then easily generate the block-encoding of $I - M_1$ by LCU, with a $1 + \alpha_A h$ normalization factor. By the analysis in [33], we know $\|I - M_1\|\|(I - M_1)^{-1}\| \leq 2k$. Using Lemma 12, we can implement a $(\mathcal{O}(k), \epsilon_1)$-block-encoding of $(I - M_1)^{-1}$ using $\mathcal{O}(k \log(1/\varepsilon_1))$ queries to the block-encoding of $I - M_1$ thus the same queries to $U_A$.

For the block-encoding of $M_2$, since it is sparse, we may directly implement it through the sparse input model. We may just assume the normalization factor to be $\sqrt{k}$ and there is no error. Then we can implement a $(\mathcal{O}(k^{3/2}), \sqrt{k}\varepsilon_1)$-block-encoding of $(I - M_1)^{-1} M_2$ using $\mathcal{O}(k \log(1/\varepsilon_1))$ queries to $U_A$ and other additional gates.

For the block-encoding for $N$, we can see the normalization factor would be the same scaling as $M_2(I - M_1)^{-1}$, thus we implemented a $(\mathcal{O}(k^{3/2}), \mathcal{O}(\sqrt{k}\varepsilon_1))$-block-encoding of $N$. $\qquad\square$

Now if we use QSVT to perform the matrix inverse, just as described in Theorem 13, we can implement an $(\alpha_{L^{-1}}, \varepsilon_2 + \frac{\alpha_{L^{-1}}^2}{\alpha_L - \varepsilon_1 \alpha_{L^{-1}}} \sqrt{k}\varepsilon_1)$-block-encoding $U_{L^{-1}}$ that encodes $L^{-1}$ in cost $\mathcal{O}(\kappa_L \log(1/\varepsilon_2))$ queries to $U_L$, i.e. $\mathcal{O}(\kappa_L \log(1/\varepsilon_2) k \log(1/\varepsilon_1))$ queries to $U_A$.

In order to perform the inverse of $L$, we need to know the condition number of $L$.

**Lemma 16** (Modified from [33], Theorem 3 & 4). *Suppose $E$ is the solution operator block-encoded by $U_{L^{-1}}$ that approximates $e^{AT}$ and $m$ is the number of steps. Let*

$$(k+1)! \geq \frac{me^3}{\delta} C_A,$$

(48)

*where*

$$\sup_t \| \exp(At)\| \leq C_A,$$

(49)

*we have*

$$\|E - e^{AT}\| \leq \delta, \quad \|L^{-1}\| = \mathcal{O}(mC_A(1+\delta))$$

(50)

*and*

$$\kappa_L \leq \mathcal{O}(m\sqrt{k}C_A(1+\delta)).$$

(51)

*Proof.* The proof can be seen at [33, Theorem 4]. $\qquad\square$

**Theorem 17.** *Let matrix $A$ be a Hurwitz matrix with $\sup_t \| \exp(tA)\|$ bounded by some constant $\rho$, and $O_A$ is a $(\alpha_A, 0)$-block-encoding of $A$. Then we can construct a $(\zeta T, \epsilon)$-block-encoding of $e^{AT}$ using*

$$\widetilde{\mathcal{O}}\left( \alpha_A \rho T \cdot \text{poly} \log \left( \frac{1}{\epsilon} \right) \right)$$

(52)

*queries to $O_A$ and same queries to other additional elementary gates. Here $\zeta = \mathcal{O}(\alpha_A \rho)$.*

*Proof.* Firstly notice that $m = \mathcal{O}(\alpha_A T)$ and $k = \mathcal{O}\left( \frac{\log(T\alpha_A \rho/\delta)}{\log \log(T\alpha_A \rho/\delta)} \right)$. For a given accuracy $\varepsilon$, we need $\delta \leq \varepsilon$, which could be given by letting

$$\varepsilon_2 + \frac{\alpha_{L^{-1}}^2}{\alpha_L - \varepsilon_1 \alpha_{L^{-1}}} \sqrt{k}\varepsilon_1 \leq \varepsilon.$$

(53)

Choose $\varepsilon_1 = \widetilde{\mathcal{O}}(\varepsilon/T^2)$ and $\varepsilon_2 = \varepsilon/2$, from the discussion above we know we can implement an $(\alpha_{L^{-1}}, \varepsilon)$-block-encoding of $e^{TA}$. Since $\alpha_{L^{-1}} = \mathcal{O}(m\rho(1+\delta))$, the queries we need is $\widetilde{\mathcal{O}}(\alpha_A \rho T \cdot \text{poly} \log(1/\varepsilon))$. $\qquad\square$

**Remark 2.** If we directly use the theorem above to prepare the state $|e^{AT}|\psi\rangle\rangle$ for some input state $|\psi\rangle$, the dependence on $T$ would be $T^2$, which does not match the optimal scaling. This is actually due to our usage of QSVT to block-encode matrices inverses. If we adopt the common technique of padding in the quantum linear system solver, we can easily improve the current scaling into $T^{3/2}$. However, for simplicity, we do not implement these standard improvements in this work, as they does not affect our end-to-end complexity result.

## D  Quantum eigenvalue transformation for linear differential equations

Here we exploit a technique known as quantum eigenvalue transformation (QEVT) by Low and Su [42] to simulate linear dynamics $\dot{x} = \mathcal{A}x$. The algorithm (QEVT) requires that the dynamics is stable under $\mathcal{A} + \mathcal{A}^\top \leq 0$, which is stronger than $\mathcal{A}$ is Hurwitz stable.

Notice that $\mathcal{A} + \mathcal{A}^\top \leq 0$ is the stability under the common 2-norm, and $\mathcal{A}$ satisfying the Lyapunov equation $\mathcal{A}X + X\mathcal{A}^\top \leq 0$ (16) indicates the stability under the inner product induced by the matrix $X$. We hence follow [42], use a sequence of polynomial functions (known as the Faber polynomials) to approximate the time-evolution operator $e^{\mathcal{A}t}$.

**Lemma 18** (Faber truncation of matrix exponentials, [42, Lemma 27])**.** *Suppose we have a matrix $A$ where its numerical range $\mathcal{W}(A) := \{\langle\psi|A|\psi\rangle\big|\|\|\psi\rangle\| = 1\}$ is enclosed by a Faber region $\mathcal{E}$ with associated conformal maps $\Phi : \mathcal{E}^c \to \mathcal{D}^c, \Psi : \mathcal{D}^c \to \mathcal{E}^c$ and Faber polynomials $F_s(z)$. Given $t > 0$, let $e^{tz} = \sum_{j=0}^{\infty} \beta_j F_j(z)$ be the Faber expansion of the complex exponential function $e^{tz}$. Assume that $\mathcal{E}$ is convex and symmetric with respect to the real axis, lying on the left half of the complex plane, then for sufficiently large $s$,*

$$\left\| e^{At} - \sum_{k=0}^{s-1} \beta_j F_j(A) \right\| = \mathcal{O}\left(\left(\frac{ct}{s}\right)^s\right), \tag{54}$$

*where $c$ is a constant determined by the conformal maps.*

By Lemma 18, given any $\varepsilon > 0$, it suffices to choose $s \sim \mathcal{O}(t \log(\frac{1}{\varepsilon}))$ such that

$$\left\| e^{At} - \sum_{k=0}^{s-1} \beta_j F_j(A) \right\| \leq \varepsilon. \tag{55}$$

**Definition 7.** For some matrix $A$ of size $N \times N$ with its normalization factor $\alpha_A$, denote the identity matrix of size $N \times N$ as $I_N$, and

$$L_N = \begin{bmatrix} 0 & & & & & \\ 1 & 0 & & & & \\ & 1 & 0 & & & \\ & & 1 & 0 & & \\ & & & \ddots & \ddots & \\ & & & & 1 & 0 \end{bmatrix}_{N \times N} \tag{56}$$

as the lower shift matrix of size $N \times N$. We then define

$$\text{PAD}(A) := |0\rangle\langle 0| \otimes \left( L_N \Psi(L_N^{-1}) \otimes I_N - L_N \otimes \frac{A}{\alpha_A} \right)$$
$$+ |1\rangle\langle 0| \otimes |0\rangle\langle N-1| \otimes (-I_N) + |1\rangle\langle 1| \otimes (I_N - L_N) \otimes I_N, \tag{57}$$

and

$$\text{PAD}(B) := |0\rangle\langle 0| \otimes \Psi'(L_N^{-1}) \otimes I_N + |1\rangle\langle 1| \otimes I_N \otimes I_N. \tag{58}$$

Note that the conformal map $\Psi$ has its Laurent expansion, and so are $\Psi'(\omega)$ and $\omega\Psi(\omega^{-1})$. Furthermore, $\Psi'(\omega)$ and $\omega\Psi(\omega^{-1})$ only contains terms with non-negative exponents. Thus $\Psi(L_N^{-1})$ and $\Psi'(L_N^{-1})$ are well-defined even though $L_N^{-1}$ is not invertible itself.

With the matrices $\text{PAD}(A)$ and $\text{PAD}(B)$, we can compute the matrix $\text{PAD}(A)^{-1}\text{PAD}(B)$, say

$$\text{PAD}(A)^{-1}\text{PAD}(B) = \left[\begin{array}{cccc:cccc} F_0(\frac{A}{\alpha_A}) & 0 & \cdots & 0 & 0 & 0 & \cdots & 0 \\ F_1(\frac{A}{\alpha_A}) & F_0(\frac{A}{\alpha_A}) & \ddots & \vdots & 0 & 0 & \cdots & 0 \\ \vdots & \ddots & \ddots & 0 & \vdots & \vdots & \ddots & \vdots \\ F_{s-1}(\frac{A}{\alpha_A}) & F_{n-2}(\frac{A}{\alpha_A}) & \cdots & F_0(\frac{A}{\alpha_A}) & 0 & 0 & \cdots & 0 \\ \hdashline F_{s-1}(\frac{A}{\alpha_A}) & F_{s-2}(\frac{A}{\alpha_A}) & \cdots & F_0(\frac{A}{\alpha_A}) & I & 0 & \cdots & 0 \\ F_{s-1}(\frac{A}{\alpha_A}) & F_{s-2}(\frac{A}{\alpha_A}) & \cdots & F_0(\frac{A}{\alpha_A}) & I & I & \cdots & 0 \\ \vdots & \vdots & \cdots & \vdots & \vdots & \vdots & \ddots & 0 \\ F_{s-1}(\frac{A}{\alpha_A}) & F_{s-2}(\frac{A}{\alpha_A}) & \cdots & F_0(\frac{A}{\alpha_A}) & I & I & \cdots & I \end{array}\right]. \quad (59)$$

Note that this matrix has $(2s) \times (2s)$ blocks, and each block of size $N \times N$.

**Definition 8.** For a matrix $A$ whose numerical range is enclosed by a Faber region $\mathcal{E}$ with associated conformal maps $\Phi : \mathcal{E}^c \to \mathcal{D}^c, \Psi : \mathcal{D}^c \to \mathcal{E}^c$ and Faber polynomials $F_s(z)$, consider some Faber expansion till $s$-th order, we then define

$$\rho \geq \max_{j=1,\cdots,s} \left\| \frac{F_j'(\frac{A}{\alpha_A})}{j} \right\| \quad (60)$$

as an upper bound on the derivative of Faber polynomials.

**Lemma 19.** *We are able to have a $(\xi, \varepsilon)$-block-encoding to $(PAD(A))^{-1}$ using $\mathcal{O}(\xi \log(\frac{1}{\varepsilon}))$ queries to the block-encoding of $A$ and $\widetilde{\mathcal{O}}(1)$ additional elementary gates. Here $\xi \sim \mathcal{O}(\rho s)$.*

*Proof.* Since we know that $\text{PAD}(A)$ can be block-encoded with an $\mathcal{O}(1)$ normalization factor with arbitrary precision using a single query to the block-encoding of $A$ and a polylogarithmic number of elementary gates, and $\|(\text{PAD}(A))^{-1}\| = \mathcal{O}(s\rho)$, the rest of the proof is done by Lemma 12. $\qquad\square$

**Theorem 20** (Modified version of Theorem 10. [42])**.** *Let matrix $A$ have its numerical range $\mathcal{W}(A) := \{\langle \psi|A|\psi\rangle \big| \|\psi\| = 1\}$ enclosed by a Faber region $\mathcal{E}$ with associated conformal maps $\Phi : \mathcal{E}^c \to \mathcal{D}^c, \Psi : \mathcal{D}^c \to \mathcal{E}^c$ and Faber polynomials $F_s(z)$. Let $p(z) = \sum_{k=0}^{s-1} \beta_k F_k(z)$ be the Faber expansion of a degree-$(s-1)$ polynomial $p$ and $O_\beta|0\rangle = \sum_{k=0}^{s-1} \beta_k |n-1-k\rangle/\|\beta\|$ be the oracle preparing the coefficients. Then, we can construct a $(\xi'\|\beta\|, \|\beta\|\varepsilon)$-block-encoding of $\sum_{k=0}^{s-1} \beta_k F_k(A/\alpha_A)$ using*

$$\mathcal{O}\left(\rho s \log\left(\frac{1}{\varepsilon}\right)\right) \quad (61)$$

*queries to $O_A$, one query to $O_\beta$ and $\widetilde{\mathcal{O}}(1)$ additional gate. Here, $\xi' \leq \mathcal{O}(\rho s)$.*

*Proof.* First, we observe that

$$(\langle 0| \otimes \langle 0| \otimes I) \left(\text{PAD}(A)^{-1}\text{PAD}(B)(X \otimes O_\beta \otimes I)\right)(|0\rangle \otimes |0\rangle \otimes I) = \sum_{k=0}^{s-1} \frac{\beta_k}{\|\beta\|} F_k(A/\alpha_A). \quad (62)$$

The $X$ is the quantum Pauli-X gate. Here the circuit has 3 registers, the first one only has one qubit, the second has $\log_2 s$ qubits, and the third one matches the size of $A$.

By [42, Theorem 9], we can implement a block-encoding of $\text{PAD}(A)$ using a single query to the block-encoded matrix $A/\alpha_A$. Also, given a constant

$$\rho \geq \max_{j=1,\cdots,s} \left\| \frac{F_j'(\frac{A}{\alpha_A})}{j} \right\|, \quad (63)$$

we know that $\|\text{PAD}(A)^{-1}\| = \mathcal{O}(\rho s)$. By Lemma 19, we can construct a $(\xi, \varepsilon)$-block-encoding of $\text{PAD}(A)^{-1}$ using $\mathcal{O}(\xi \log(1/\epsilon))$ queries to the block-encoding of $A/\alpha_A$.

Also, assume we can implement a block-encoding of $\text{PAD}(B)$ with a $\mathcal{O}(1)$ normalization factor. Combining these two block-encodings together, we obtain a $(\xi', \varepsilon)$-block-encoding of $\text{PAD}(A)^{-1}\text{PAD}(B)$, where $\xi' \leq \mathcal{O}(\rho s)$. Then, by (62), the block-encoding we implemented turns out to be a $(\xi'\|\beta\|, \|\beta\|\varepsilon)$-block-encoding of $\sum_{k=0}^{s-1} \beta_k F_k(A/\alpha_A)$. $\qquad \square$

**Theorem 21** (Block-encoding of matrix exponential). *Suppose that $A \in \mathbb{R}^{m \times n}$ is a Hurwitz matrix, and $O_A$ is a $(\alpha_A, 0)$-block-encoding of $A$. Then we can implement a $(\zeta t, \varepsilon)$-block-encoding of $e^{At}$ using*

$$\mathcal{O}\left(\alpha_A t \operatorname{poly}\log\left(\frac{1}{\varepsilon}\right)\right) \tag{64}$$

*queries to $O_A$, one query to a state preparation oracle $O_\beta$ and $\widetilde{\mathcal{O}}(1)$ additional gate. Here $\zeta = \mathcal{O}(\alpha_A \operatorname{poly}\log(1/\epsilon))$.*

*Proof.* First we use Lemma 18 to $e^{\alpha_A (A/\alpha_A) t}$, then we know the polynomial should be of degree $\mathcal{O}(\alpha_A t \log(\frac{1}{\varepsilon_1}))$ in order to have a $\varepsilon_1$ accuracy. Then leverage Theorem 20, we need $\mathcal{O}\left(\rho\left(\alpha_A t \log\left(\frac{1}{\varepsilon_1}\right)\right) \log\left(\frac{1}{\varepsilon_2}\right)\right)$ queries to $O_A$ to construct a block-encoding of the polynomial. Since the Faber polynomial expansion of $e^{At}$ converges absolutely, without loss of generality, we assume $\|\beta\|$ is $\mathcal{O}(1)$. Choosing $\varepsilon_1 = \frac{\varepsilon}{2}$ and $\varepsilon_2 = \frac{\varepsilon}{2\|\beta\|}$, we have a $(\zeta t, \varepsilon)$-block-encoding of $e^{At}$. $\qquad \square$

# E   Proof of Theorem 4

**Lemma 22** (Select oracle). *Suppose that $\mathcal{A}$ is Hurwitz, and $O_{\mathcal{A}}$ is an $(\alpha, 0)$-block-encoding of $\mathcal{A}$. Let $K$ be a positive integer and $\tau, \varepsilon > 0$ are two real-valued scalars. The select oracle as defined in Definition 4 can be implemented using*

$$\widetilde{\mathcal{O}}\left(\alpha \rho \tau K \log(K) \cdot \operatorname{poly}\log\left(\frac{1}{\varepsilon}\right)\right) \tag{65}$$

*queries to controlled $O_{\mathcal{A}}$, and $\widetilde{\mathcal{O}}\left(\alpha \rho \tau K \log(K) \cdot \operatorname{poly}\log\left(\frac{1}{\varepsilon}\right)\right)$ queries to other additional elementary gates.*

*Proof.* Firstly, $t_k$ here attains $t_k = \frac{k\tau}{K}$. By Theorem 17, for each $k = 0, \ldots, K$, we can implement a $(\zeta t_k, \varepsilon)$-block-encoding of $e^{At_k}$ using $\widetilde{\mathcal{O}}(\alpha \rho t_k \cdot \operatorname{poly}\log(1/\varepsilon))$ queries to $O_{\mathcal{A}}$. We denote this block-encoding as $U_k$. Notice that the select oracle $\operatorname{select}(\mathcal{A}, \varepsilon)$ can be represented by

$$\operatorname{select}(\mathcal{A}, \varepsilon) = \prod_{k=0}^{K} \left[(\mathbb{I} - |k\rangle\langle k|) \otimes \mathbb{I} + |k\rangle\langle k| \otimes U_k\right], \tag{66}$$

where each unitary operator $[(\mathbb{I} - |k\rangle\langle k|) \otimes \mathbb{I} + |k\rangle\langle k| \otimes U_k]$ in this product is a $k$-controlled version of $U_k$ that can be implemented with $\widetilde{\mathcal{O}}(\alpha \rho t_k \log(K) \cdot \operatorname{poly}\log(1/\varepsilon))$ queries to the controlled $O_{\mathcal{A}}$ and other additional gates. Therefore, the overall query complexity is

$$\widetilde{\mathcal{O}}\left(\alpha \rho \tau \left(\sum_{k=0}^{K} t_k\right) \log(K) \cdot \operatorname{poly}\log\left(\frac{1}{\varepsilon}\right)\right) \leq \widetilde{\mathcal{O}}\left(\alpha \rho \tau K \log(K) \cdot \operatorname{poly}\log\left(\frac{1}{\varepsilon}\right)\right), \tag{67}$$

where the last step follows from $\sum_{k=0}^{K} t_k = \sum_{k=0}^{K} \frac{k\tau}{K} = \frac{(K+1)\tau}{2}$. $\qquad \square$

In the following proof, we will need a state preparation oracle

$$O_\lambda |0\rangle = \frac{1}{\sqrt{\|\lambda\|_1}} \sum_{k=0}^{K} \sqrt{\lambda_k} |k\rangle, \tag{68}$$

where $\lambda_k = w_k k^2$, $\|\lambda\|_1 = \sum_{k=0}^{K} |\lambda_k|$ and $w_k = \frac{(2-1_{k=0,K})\tau}{2K}$. Since $\lambda_k$ can be expressed as a smooth function of $k$ for $k \in \{0, 1, \cdots, K\}$, this state preparation oracle can be implemented in $\mathcal{O}(\text{poly}\log(K))$ cost.

Now, we are ready to prove Theorem 4.

*Proof.* The global error of the trapezoidal rule (see (20)) is proportional to

$$\max_{\xi \in [0,\tau]} \frac{\tau^3}{K^2} |F''(\xi)|, \tag{69}$$

where

$$|F''(\xi)| = \left| \mathcal{A}^2 F(\xi) + 2\mathcal{A}F(\xi)A^\top + F(\xi)(\mathcal{A}^\top)^2 \right| \leq 4\|\mathcal{A}\|^2 \|\Omega\|.$$

Therefore, to achieve precision $\varepsilon_1$, the total number of quadrature points is

$$K = \mathcal{O}\left( \frac{\tau^{3/2} \alpha \eta^{1/2}}{\varepsilon_1^{1/2}} \right). \tag{70}$$

Now, we consider the following two select oracles, as defined in Definition 4:

$$\text{select}(\mathcal{A}, \varepsilon_2) := \sum_{k=0}^{K} |k\rangle\langle k| \otimes U_k, \quad \text{select}(\mathcal{A}^\top, \varepsilon_2) := \sum_{k=0}^{K} |k\rangle\langle k| \otimes V_k, \tag{71}$$

where $U_k$ (or $V_k$) is a $(\zeta t_k, \varepsilon_2)$-block-encoding of $e^{At_k}$ (or $e^{A^\top t_k}$) for $0 \leq k \leq K$. Recall that $O_\Omega$ is a $(\eta, 0)$-block-encoding of $\Omega$, it turns out that

$$\text{select}(\mathcal{A}, \varepsilon)(I \otimes O_\Omega)\text{select}(\mathcal{A}^\top, \varepsilon) = \sum_{k=0}^{K} |k\rangle\langle k| \otimes W_k, \tag{72}$$

where $W_k := U_k U_\Omega V_k$ is a $(\eta \zeta^2 t_k^2, 2\zeta t_k \eta \varepsilon_2)$-block-encoding of the matrix $\mathscr{F}(t_k)$.

Let $O_\lambda$ be the state preparation oracle defined in (68). By the LCU technique [22, Lemma 52], we can implement a $(\gamma, \sum_{k=0}^{K} 2|w_k|\eta \zeta t_k \varepsilon_2)$-block-encoding (where $\gamma := \sum_{k=0}^{K} |w_k| \zeta^2 t_k^2 \eta$) of $\sum_{k=0}^{K} w_k e^{At_k} \Omega e^{A^\top t_k}$ using a single query to $\text{select}(A, \varepsilon_2)$, $\text{select}(A^\top, \varepsilon_2)$, $U_\Omega$, $O_\lambda$ and its inverse.

Notice that

$$\sum_{k=0}^{K} |w_k| t_k \leq \sum_{k=0}^{K} \frac{\tau}{K} \frac{k\tau}{K} = \mathcal{O}(\tau^2), \tag{73}$$

thus

$$\sum_{k=0}^{K} 2|w_k|\eta \zeta t_k \varepsilon_2 = \widetilde{\mathcal{O}}\left( \tau^2 \eta \zeta \varepsilon_2 \right). \tag{74}$$

Given a positive scalar $\varepsilon > 0$, and set the integration time

$$\tau = \kappa \log\left( \frac{3\|X^*\|\kappa}{\varepsilon\|w\|_1 \lambda_{\min}(X^*)} \right), \tag{75}$$

Note that

$$\varepsilon^\delta \text{poly}\left( \log\left( \frac{1}{\varepsilon} \right) \right) \to 0, \tag{76}$$

then if we choose

$$\varepsilon_1 = \frac{\varepsilon}{3}, \quad \varepsilon_2 = \mathcal{O}(\varepsilon^{1+\delta}), \tag{77}$$

where $\delta > 0$ is a constant positive number. By Lemma 3 and the error analysis above, we implement a $(\gamma, \varepsilon)$-block-encoding of $X^*$, with

$$\gamma = \sum_{k=0}^{K} |w_k| \zeta^2 t_k^2 \eta = \mathcal{O}(\zeta^2 \eta \tau^3) = \widetilde{\mathcal{O}}(\alpha^2 \rho^2 \kappa^3 \eta). \tag{78}$$

It follows that the overall cost of this block-encoding is

$$\widetilde{\mathcal{O}}\left(\frac{\alpha^2 \rho \kappa^{5/2} \eta^{1/2}}{\varepsilon^{1/2}}\right) \tag{79}$$

queries to $O_{\mathcal{A}}$ and same queries to other additional elementary gates. $\qquad\square$

## F  Quantum trace estimation

Here we show how to estimate the trace of a positive definite block-encoded matrix. The main result is in Theorem 25.

**Lemma 23.** *Let $U_H$ be an $(\alpha, m, \varepsilon_H)$-block-encoding of a positive-definite Hermitian matrix $H \in \mathbb{C}^{n \times n}$ and let $\varepsilon \in (0, \frac{1}{3}]$. Let $\lambda_{\min} > 0$ be a known lower bound on the smallest eigenvalue of $H$. Provided that $\varepsilon_H \leq \mathcal{O}\left(\frac{\lambda_{\min}\varepsilon^2}{\log(1/\varepsilon)}\right)$ is sufficiently small (where $\varepsilon > 0$ is a given number), we can construct a $(1, m+2, \varepsilon)$-block-encoding of $\sqrt{\frac{H/\alpha}{36}}$ using $\mathcal{O}\left(\frac{\alpha}{\lambda_{\min}}\log(1/\varepsilon)\right)$ applications of $U_H$ and $U_H^\dagger$, a single application of controlled-$U_H$, and $\mathcal{O}\left(\frac{\alpha(m+1)}{\lambda_{\min}}\log(1/\varepsilon)\right)$ other one- and two-qubit gates. The description of this block-encoding circuit can be computed classically in time $\mathcal{O}\left(\mathrm{poly}\left(\frac{\alpha}{\lambda_{\min}}\log(1/\varepsilon)\right)\right)$.*

*Proof.* Define $\widetilde{\lambda}_{\min} := \lambda_{\min}/\alpha$, such that $\widetilde{\lambda}_{\min} I \preceq H/\alpha \preceq I$. We first find a polynomial approximation of $f(x) := \sqrt{\frac{x}{36}}$ over $x \in [\widetilde{\lambda}_{\min}, 1]$ using [22, Corollary 66]. To use this corollary, let $x_0 := 1$, $r := 1 - \widetilde{\lambda}_{\min}$, $\delta := \widetilde{\lambda}_{\min}$, and observe that $f(x_0 + x) = \frac{1}{6}\sqrt{1+x} = \frac{1}{6}\sum_{\ell=0}^{\infty}\binom{1/2}{\ell}x^\ell$ whenever $|x| \leq r + \delta = 1$. Also note that $\frac{1}{6}\sum_{\ell=0}^{\infty}\left|\binom{1/2}{\ell}\right| \leq \frac{1}{3} =: B$. Then there is an efficiently computable polynomial $P_0 \in \mathbb{C}[x]$ of degree $d = \mathcal{O}\left(\frac{1}{\widetilde{\lambda}_{\min}}\log(1/\varepsilon)\right)$ such that $\|f(x) - P_0(x)\| \leq \frac{\varepsilon}{2}$ for $x \in [\widetilde{\lambda}_{\min}, 1]$ and $\|P_0(x)\| \leq \frac{\varepsilon}{2} + B \leq \frac{1}{2}$ for $x \in [-1, 1]$. It follows that defining $P(x) := \mathrm{Re}(P_0(x))$ gives $|f(x) - P(x)|_{[\widetilde{\lambda}_{\min}, 1]} \leq \frac{\varepsilon}{2}$ and $|P(x)|_{[-1,1]} \leq \frac{1}{2}$.

Next, we use [22, Theorem 56] to construct construct a unitary $V$ that is a $(1, m+2, \varepsilon/2)$-encoding of $P(H/\alpha)$ with the desired complexity, using the promise that $\varepsilon_H$ is sufficiently small to satisfy $4d\sqrt{\varepsilon_H/\alpha} \leq \varepsilon/4$. $V$ is then a $(1, m+2, \varepsilon)$-block-encoding of $\sqrt{\frac{H/\alpha}{36}}$ since

$$
\begin{aligned}
&\left\|\sqrt{\frac{H/\alpha}{36}} - (\langle 0|^{\otimes m+2} \otimes I)V(|0\rangle^{\otimes m+2} \otimes I)\right\| \\
&\leq \left\|\sqrt{\frac{H/\alpha}{36}} - P(H/\alpha)\right\| + \left\|P(H/\alpha) - (\langle 0|^{\otimes m+2} \otimes I)V(|0\rangle^{\otimes m+2} \otimes I)\right\| \\
&\leq \|g(H/\alpha) - P(H/\alpha)\| + \frac{\varepsilon}{2} = \|\mathrm{diag}(g(\lambda_H/\alpha)) - \mathrm{diag}(P(\lambda_H/\alpha))\| + \frac{\varepsilon}{2} \\
&= \max_{\lambda_i \in \lambda_H} |g(\lambda_i/\alpha) - P(\lambda_i/\alpha)| + \frac{\varepsilon}{2} \leq \frac{\varepsilon}{2} + \frac{\varepsilon}{2} = \varepsilon,
\end{aligned}
\tag{80}
$$

where $\lambda_H \in \mathbb{R}^n$ is the vector of eigenvalues of $H$. $\qquad\square$

**Lemma 24.** *Let $\mu > 0$. Let $A$ be an $n$-by-$n$ Hermitian matrix such that $0 \preceq A \preceq I$ and let $\widetilde{V}$ be a $(1, m, \mu/3)$-block-encoding of $\sqrt{A}$. Then there exists $\widetilde{U}_A$ such that $\left|\left\|(\langle 0| \otimes I)\widetilde{U}_A|0\ldots0\rangle\right\|^2 - \frac{\mathrm{Tr}(A)}{n}\right| \leq \mu$ that uses 1 query to $\widetilde{V}$ and $\mathcal{O}(\log n)$ additional gates.*

*Proof.* Our proof is similar to the proof of [55, Lemma 13]. The idea is to first prepare the maximally entangled state $\sum_{i=1}^{n}|i\rangle|i\rangle/\sqrt{n}$ (which requires $\mathcal{O}(\log n)$ gates) and then apply the map $\sqrt{A}$ to the first register. We can assume without loss of generality that $\mu \leq 1$, otherwise the statement is trivial.

Note that $\widetilde{V}_0 := (\langle 0|^{\otimes m} \otimes I)\widetilde{V}(|0\rangle^{\otimes m} \otimes I)$ is a $\mu/3$-approximation of $\sqrt{A}$ by definition. We are interested in the probability $p$ of measuring $0^{\otimes m}$ in the first register after applying $\widetilde{V}$.

$$p := \left\| (\langle 0|^{\otimes m} \otimes I)\widetilde{V}(|0\rangle^{\otimes m} \otimes I) \sum_{i=1}^{n} \frac{|i\rangle|i\rangle}{\sqrt{n}} \right\|^2 = \left\| \widetilde{V}_0 \sum_{i=1}^{n} \frac{|i\rangle|i\rangle}{\sqrt{n}} \right\|^2$$
$$= \frac{1}{n} \sum_{i=1}^{n} \langle i|\widetilde{V}_0^\dagger \widetilde{V}_0|i\rangle = \frac{1}{n} \operatorname{Tr}\left(\widetilde{V}_0^\dagger \widetilde{V}_0\right). \tag{81}$$

It remains to show that $p$ is a good approximation of $\operatorname{Tr}(A)/n$. For this, we show $\widetilde{V}_0^\dagger \widetilde{V}_0 \approx A$. Note that for all matrices $B, \widetilde{B}$ with $\|B\| \le 1$, we have

$$\left\| B^\dagger B - \widetilde{B}^\dagger \widetilde{B} \right\| = \left\| (B^\dagger - \widetilde{B}^\dagger)B + B^\dagger(B - \widetilde{B}) - (B^\dagger - \widetilde{B}^\dagger)(B - \widetilde{B}) \right\|$$
$$\le \left\| (B^\dagger - \widetilde{B}^\dagger)B \right\| + \left\| B^\dagger(B - \widetilde{B}) \right\| + \left\| (B^\dagger - \widetilde{B}^\dagger)(B - \widetilde{B}) \right\|$$
$$\le \left\| B^\dagger - \widetilde{B}^\dagger \right\| \|B\| + \left\| B^\dagger \right\| \left\| B - \widetilde{B} \right\| + \left\| B^\dagger - \widetilde{B}^\dagger \right\| \left\| B - \widetilde{B} \right\|$$
$$\le 2 \left\| B - \widetilde{B} \right\| + \left\| B - \widetilde{B} \right\|^2. \tag{82}$$

Using equation (81) along with equation (82) (letting $B = \sqrt{A}$ and $\widetilde{B} = \widetilde{V}_0$), we see

$$\left| \frac{\operatorname{Tr}(A)}{n} - p \right| = \frac{1}{n} \left| \operatorname{Tr}\left(A - \widetilde{V}_0^\dagger \widetilde{V}_0\right) \right| \le \left\| A - \widetilde{V}_0^\dagger \widetilde{V}_0 \right\| \le 2\left(\frac{\mu}{3}\right) + \left(\frac{\mu}{3}\right)^2 \le \mu. \tag{83}$$

$\square$

**Theorem 25.** *Let $U_H$ be an $(\alpha, m, \varepsilon)$-block-encoding of a Hermitian matrix $H \in \mathbb{C}^{n \times n}$, where $U_H$ can be implemented using $T_H$ elementary gates. Suppose $H \succ 0$ and let $\lambda_{\min} > 0$ be a known lower bound on the smallest eigenvalue of $H$. Then, with probability at least $4/5$, we can estimate $\operatorname{Tr}(H)$ to within multiplicative error $\theta \in (0, 1]$ in cost*

$$\mathcal{O}\left( \frac{\alpha^{3/2} T_H}{\theta \lambda_{\min}^{3/2}} \log\left(\frac{\alpha}{\theta \lambda_{\min}}\right) \right), \tag{84}$$

*provided that $\varepsilon \le \mathcal{O}\left( \frac{\lambda_{\min}^3 \theta^2}{\alpha^2 \log(\alpha/\theta \lambda_{\min})} \right)$.*

*Proof.* Our proof is similar to the proof of [55, Corollary 10]. Let $A := \frac{H/\alpha}{36}$ and note that since $H/\alpha \succeq \lambda_{\min} I/\alpha$, we have

$$\frac{\operatorname{Tr}(A)}{n} := \frac{\operatorname{Tr}(H/\alpha)}{36n} \ge \frac{\lambda_{\min}}{36\alpha}. \tag{85}$$

Let $\widetilde{U}_A$ be a $(1, m + 2, \theta\lambda_{\min}/216\alpha)$-block-encoding of $\sqrt{A}$ constructed via Lemma 23. Using Lemma 24, we can construct a unitary circuit $\widetilde{V}$ such that

$$\left| \left\| (\langle 0| \otimes I)\widetilde{V}|0\ldots 0\rangle \right\|^2 - \frac{\operatorname{Tr}(A)}{n} \right| \le \frac{\theta\lambda_{\min}}{72\alpha} \le \frac{\theta}{2} \cdot \frac{\operatorname{Tr}(A)}{n} \le \frac{\operatorname{Tr}(A)}{2n} \tag{86}$$

Therefore we have

$$\left\| (\langle 0| \otimes I)\widetilde{V}|0\ldots 0\rangle \right\|^2 \ge \frac{\operatorname{Tr}(A)}{n} - \frac{\operatorname{Tr}(A)}{2n} = \frac{\operatorname{Tr}(A)}{2n} \ge \frac{\lambda_{\min}}{72\alpha} =: p_{\min}. \tag{87}$$

as well as

$$\left\| (\langle 0| \otimes I)\widetilde{V}|0\ldots 0\rangle \right\|^2 \le \frac{\operatorname{Tr}(A)}{n} + \frac{\operatorname{Tr}(A)}{2n} = \frac{3\operatorname{Tr}(A)}{2n}. \tag{88}$$

Using the amplitude estimation technique of [55, Lemma 9] with $p_{\min} = \lambda_{\min}/72\alpha$ and $\mu = \theta/3$ gives us a $\widetilde{p}$ that with probability at least $4/5$ satisfies

$$\left| \widetilde{p} - \left\| (\langle 0| \otimes I)\widetilde{V}|0\ldots 0\rangle \right\|^2 \right| \le \frac{\theta}{3} \left\| (\langle 0| \otimes I)\widetilde{V}|0\ldots 0\rangle \right\|^2 \le \frac{\theta}{2} \cdot \frac{\operatorname{Tr}(A)}{n}. \tag{89}$$

By combining (86) and (89) and using the triangle inequality, we get

$$\left| \frac{\text{Tr}(A)}{n} - \widetilde{p} \right| \leq \theta \cdot \frac{\text{Tr}(A)}{n}. \tag{90}$$

Therefore, $36\alpha n \widetilde{p}$ is a multiplicative $\theta$-approximation of $\text{Tr}(H)$.

It remains to show that the complexity statement holds. The construction of $\widetilde{U}_A$ via Lemma 23 uses $\mathcal{O}(T_H \cdot \frac{\alpha}{\lambda_{\min}} \log(\alpha/\theta\lambda_{\min}))$ elementary gates from the applications of $U_H$, $U_H^\dagger$, and controlled-$U_H$, and $\mathcal{O}((m+1)\frac{\alpha}{\lambda_{\min}} \log(\alpha/\theta\lambda_{\min}))$ other one- and two-qubit gates. The construction of $\widetilde{V}$ requires an additional $\mathcal{O}(\log n)$ gates. The amplitude estimation step requires $\mathcal{O}(\sqrt{\alpha/\lambda_{\min}}/\theta)$ uses of $\widetilde{V}$ and $\widetilde{V}^\dagger$, and $\mathcal{O}(\sqrt{\alpha/\lambda_{\min}} \log n/\theta)$ additional gates. So the overall complexity is

$$\mathcal{O}\left( \frac{\sqrt{\alpha/\lambda_{\min}}}{\theta} \left( \frac{\alpha T_H}{\lambda_{\min}} \log\left( \frac{\alpha}{\theta\lambda_{\min}} \right) + \frac{(m+1)\alpha}{\lambda_{\min}} \log\left( \frac{\alpha}{\theta\lambda_{\min}} \right) + \log n \right) \right). \tag{91}$$

Noting that $T_H \geq m$ and $T_H \geq \log n$ for any non-trivial $U_H$ gives the stated bound. $\qquad\square$

## G  Quantum gradient estimation

**Lemma 26.** *Given $K \in \mathcal{S}_K(a)$ and $\|K - K^*\|_F > \varepsilon$, we have*

$$\|\nabla f\| \geq c\varepsilon, \tag{92}$$

*where the constant $c = \sqrt{2\mu_f \nu \lambda_{\min}(R)/a}$.*

*Proof.* This lemma follows directly from Lemma 9 and Lemma 10. $\qquad\square$

**Lemma 27** (Block-encoded $\nabla f(K)$)**.** *Assume that we have efficient procedures (as described in Assumption 1) to access the problem data $A, B, Q, R$ in $\mathcal{O}(\text{poly}\log(n))$ time. Let $K \in \mathcal{S}_K(a)$ be a stabilizing policy stored in a quantum-accessible data structure. Given any $\varepsilon_b > 0$, we can implement a $(\gamma_\nabla, \varepsilon_b)$-block-encoding of $\nabla f(K)$ in cost $\widetilde{\mathcal{O}}\left( a^6 \rho^3 \sqrt{\frac{\kappa^{11}}{\varepsilon_b}} \right)$, where $\gamma_\nabla \leq \widetilde{\mathcal{O}}(a^6 \rho^4 \kappa^6)$.*

*Proof.* We know that for the function $f(K)$ defined in (5), the gradient has a closed-form expression: $\nabla f(K) = 2(RK - B^\top P(K))X(K)$. As shown in the proof of Theorem 5, we can implement a $(\gamma_P, \varepsilon)$-block-encoding of $P(K)$ in cost $\widetilde{\mathcal{O}}\left( a^3 \rho \sqrt{\frac{\kappa^5}{\varepsilon}} \right)$, where $\gamma_P \leq \widetilde{\mathcal{O}}(a^4 \rho^2 \kappa^3)$. Similarly, since $X(K)$ is the solution to the Lyapunov equation (assuming $\Sigma_0 = \mathbb{I}$)

$$(A - BK)X + X(A - BK)^\top + \mathbb{I} = 0, \tag{93}$$

we can implement a $(\gamma_X, \varepsilon)$-block-encoding of $X(K)$ in cost $\widetilde{\mathcal{O}}\left( a^2 \rho \sqrt{\frac{\kappa^5}{\varepsilon}} \right)$, where $\gamma_X \leq \widetilde{\mathcal{O}}(a^2 \rho^2 \kappa^3)$. Based on our assumptions on the input procedures and the usage of quantum data structure, we can implement a $(s\|K\|_F, 0)$-block-encoding of $RK$ and a $(s\gamma_P, s\varepsilon)$-block-encoding of $B^\top P(K)$. It follows that a $(\gamma_\nabla, \varepsilon_b)$-block-encoding of $\nabla f(K)$ can be implemented in cost $\widetilde{\mathcal{O}}\left( a^3 \rho \sqrt{\frac{\kappa^5}{\varepsilon}} \right)$, where

$$\gamma_\nabla := \frac{1}{2} s\gamma_X(\gamma_P + \|K\|_F) \leq \widetilde{O}\left( a^6 \rho^4 \kappa^6 \right), \tag{94}$$

and $\varepsilon_b \leq \mathcal{O}((a + \gamma_P + \gamma_X\gamma_P)\varepsilon)$. To achieve the desired precision, we pass the error parameter $\varepsilon \to \varepsilon/(2(a + \gamma_P + \gamma_X\gamma_P))$ to the asymptotic cost. $\qquad\square$

**Definition 9.** For a matrix $G$ of size $m \times n$, we could always write is as
$$G = [G_1 \quad G_2 \quad \cdots \quad G_n], \tag{95}$$
where $G_i$ is a column vector of size $m \times 1$. Define $|G_{\mathrm{vec}}\rangle$ as
$$|G_{\mathrm{vec}}\rangle = \begin{bmatrix} G_1 \\ G_2 \\ \vdots \\ G_n \end{bmatrix} / \|G\|_F. \tag{96}$$

**Lemma 28** (Matrix Vectorization). *Let $U_G$ be a $(\gamma_\nabla, \varepsilon)$-block-encoding of $(\nabla f(K))^\top$, and we denote*
$$G := \gamma_\nabla(\langle 0^r| \otimes I)U_G(|0^r\rangle \otimes I), \tag{97}$$
*where $r$ is the number of ancilla qubits required. Then, we can prepare a quantum state $|G_{\mathrm{vec}}\rangle$ with $\Omega(1)$ success probability using $\widetilde{\mathcal{O}}\left(\frac{\gamma_\nabla \sqrt{m}}{\|G\|_F}\right)$ queries to $U_G$ and its inverse.*

*Proof.* First, from Lemma 27 we know that we have $U_G^\top$ that encodes $\nabla f(K)^\top$. Notice that $\nabla f(K)$ is of size $m \times n$, so

$$I^{r_1} \otimes (U_G^{r_2, r_3})^\top \left( \frac{1}{\sqrt{m}} \sum_{i=0}^{m-1} |i\rangle^{r_1} \otimes (|0^a\rangle)^{r_2} \otimes |i\rangle^{r_3} \right)$$
$$= \frac{1}{\gamma_\nabla \sqrt{m}} \sum_{i=0}^{m-1} |i\rangle^{r_1} \otimes (|0^a\rangle)^{r_2} \otimes (G^\top |i\rangle)^{r_3} + |\perp\rangle \tag{98}$$
$$= \frac{\|G\|_F}{\gamma_\nabla \sqrt{m}} \frac{\sum_{i=0}^{m-1} |i\rangle^{r_1} \otimes (|0^a\rangle)^{r_2} \otimes (G^\top |i\rangle)^{r_3}}{\|G\|_F} + |\perp\rangle$$

where $r_1, r_2, r_3$ are the indexes for registers, where the $r_1$ register has $\log_2(m)$ qubits and the $r_3$ register has $\log_2(n)$ qubits(We can always assume $m$ and $n$ are the power of 2). The state $|\perp\rangle$ contains the state satisfy
$$(I^{r_1} \otimes \langle 0^a|^{r_2} \otimes I^{r_3})|\perp\rangle = 0. \tag{99}$$

By leveraging amplitude amplification, we can get the state $|G_{\mathrm{vec}}\rangle$ using $\mathcal{O}(\frac{\gamma_\nabla \sqrt{m}}{\|G\|_F})$ queries to the block-encoding $U_G^\top$. $\qquad\square$

**Lemma 29** (Gradient entry estimation). *Denote the state preparation in Lemma 28 for $|G_{vec}\rangle$ as $U_{G_{vec}}$, we are able to get the classical $\mathcal{G}$ with*
$$\|\mathcal{G} - |G_{\mathrm{vec}}\rangle\| \le \varepsilon_r \tag{100}$$
*using*
$$\widetilde{\mathcal{O}}\left( \frac{\gamma_\nabla}{\|G\|_F} \frac{m^{3/2}n}{\varepsilon_r} \right) \tag{101}$$
*queries to $U_G$ and $\widetilde{\mathcal{O}}\left( \frac{\gamma_\nabla}{\|G\|_F} \frac{m^{3/2}n}{\varepsilon_r} \right)$ additional elementary gates.*

*Proof.* [7, Theorem 2] indicates that we can get $\mathcal{G}$ using $\widetilde{\mathcal{O}}(\frac{mn}{\epsilon_r})$ queries to the oracle that prepares $|G_{\mathrm{vec}}\rangle$ and $\widetilde{\mathcal{O}}\left( \frac{mn}{\varepsilon_r} \right)$ additional elementary gates. Since each $U_{G_{\mathrm{vec}}}$ requires
$$\widetilde{\mathcal{O}}\left( \frac{\gamma_\nabla \sqrt{m}}{\|G\|_F} \right) \tag{102}$$
queries, we know the total cost should be
$$\widetilde{\mathcal{O}}\left( \frac{\gamma_\nabla}{\|G\|_F} \frac{m^{3/2}n}{\varepsilon_r} \right). \tag{103}$$

$\square$

**Lemma 30** (Gradient norm estimation). *Suppose we have the block-encoding $U_G$ defined in Lemma 27, where*

$$G = \gamma_\nabla(\langle 0 | \otimes I)U_G(|0\rangle \otimes I), \tag{104}$$

*then we are able to estimate $\|G\|_F$ up an additive error $\varepsilon_a$ using*

$$\mathcal{O}\left(\frac{\gamma_\nabla \sqrt{m}\|G\|_F}{\varepsilon_a^2 + 2\|G\|_F \varepsilon_a}\right) \tag{105}$$

*queries to $U_G$ and its inverse and also $\widetilde{\mathcal{O}}\left(\frac{\gamma_\nabla \sqrt{m}\|G\|_F}{\varepsilon_a^2 + 2\|G\|_F \varepsilon_a}\right)$ elementary gates.*

*Proof.* We still need the computation we did in equation 98, say

$$I^{r_1} \otimes (U_G^{r_2,r_3})^\top \left(\frac{1}{\sqrt{m}} \sum_{i=0}^{m-1} |i\rangle^{r_1} \otimes (|0^a\rangle)^{r_2} \otimes |i\rangle^{r_3}\right)$$
$$= \frac{\|G\|_F}{\gamma_\nabla \sqrt{m}} \frac{\sum_{i=0}^{m-1} |i\rangle^{r_1} \otimes (|0^a\rangle)^{r_2} \otimes (G^\top |i\rangle)^{r_3}}{\|G\|_F} + |\perp\rangle. \tag{106}$$

According to [55, Lemma 9], we can get the estimation to $\left(\frac{\|G\|_F}{\gamma_\nabla \sqrt{m}}\right)^2$ with multiplicative error $\mu$ using $\mathcal{O}(\frac{\gamma_\nabla \sqrt{m}}{\mu\|G\|_F})$ queries to $U_G^\top$. Note our estimator to $\|G\|_F$ as $a_{\text{est}}$, the goal we want to achieve is

$$|a_{\text{est}} - \|G\|_F| \leq \varepsilon_a. \tag{107}$$

Consider our estimation to the amplitude as $\left(\frac{a_{\text{est}}}{\gamma_\nabla \sqrt{m}}\right)^2$, set $\mu = \frac{\epsilon_a^2 + 2\epsilon_a \|G\|_F}{\|G\|_F^2}$, we have

$$\left(\frac{1}{\gamma_\nabla \sqrt{m}}\right)^2 |a_{\text{est}}^2 - \|G\|_F^2| = \left(\frac{1}{\gamma_\nabla \sqrt{m}}\right)^2 \left(|a_{\text{est}} - \|G\|_F| \cdot |a_{\text{est}} + \|G\|_F|\right)$$
$$\leq \left(\frac{1}{\gamma_\nabla \sqrt{m}}\right)^2 (\varepsilon_a (|a_{\text{est}} - \|G\|_F| + 2\|G\|_F)) \leq \left(\frac{1}{\gamma_\nabla \sqrt{m}}\right)^2 (\varepsilon_a^2 + 2\varepsilon_a \|G\|_F) \tag{108}$$
$$\leq \mu \left(\frac{\|G\|_F}{\gamma_\nabla \sqrt{m}}\right)^2.$$

This in turn gives us the estimation in 107. And this estimation requires

$$\mathcal{O}\left(\frac{\gamma_\nabla \sqrt{m}\|G\|_F}{\varepsilon_a^2 + 2\|G\|_F \varepsilon_a}\right) \tag{109}$$

queries to $U_G$ and its inverse. The gate complexity also follows from [55, Lemma 9]. $\qquad \square$

**Theorem 31** (Quantum gradient estimation). *Assume that we have efficient procedures (as described in Assumption 1) to access the problem data $A, B, Q, R$ in $\mathcal{O}(\text{poly}\log(n))$ time. Let $K \in \mathcal{S}_K(a)$ be a stabilizing policy stored in a quantum-accessible data structure. Provided that $\|K - K^*\| > \varepsilon$, we can compute a $\theta$-robust estimate of $\nabla f(K)$ in cost*

$$\widetilde{\mathcal{O}}\left(\frac{m^{1.5} n}{\theta^{1.5} \varepsilon^{1.5}}\right). \tag{110}$$

*Proof.* We need an estimation to $\nabla f(K)$ up to a multiplicative error $\theta$. This requires a few steps as below:

**Block-Encoding.** From Lemma 27, we know we are able to construct a $(\gamma_\nabla, \varepsilon_b)$-block-encoding to $\nabla f(K)$ using

$$\widetilde{\mathcal{O}}\left(a^6 \rho^3 \sqrt{\frac{\kappa^{11}}{\varepsilon_b}}\right) \tag{111}$$

elementary gates. And we also have the estimation that $\gamma_\nabla \leq \widetilde{\mathcal{O}}(a^6 \rho^4 \kappa^6)$.

**Entry Retrieving.** The next step is to retrieve the entries in the estimation to $\nabla f(K)$. We firstly use the technique introduced in Lemma 28 to get a state of the estimation to $|G_{\text{vec}}\rangle$, and this step requires

$$\mathcal{O}\left(\frac{\gamma_\nabla \sqrt{m}}{\|G\|_F}\right) \tag{112}$$

queries. Now we can use Lemma 29 to get the entries in $|G_{\text{vec}}\rangle$ up to an additive error $\epsilon_r$, using

$$\widetilde{\mathcal{O}}\left(\frac{\gamma_\nabla}{\|G\|_F} \frac{m^{3/2}n}{\varepsilon_r}\right) \tag{113}$$

queries to $U_G$.

**Norm Estimation.** The final step is to estimate the norm of $G$. Lemma 30 tells us that we can achieve an additive error $\varepsilon_a$ to $\|G\|_F$ using

$$\mathcal{O}\left(\frac{\gamma_\nabla \sqrt{m}\|G\|_F}{\varepsilon_a^2 + 2\|G\|_F\varepsilon_a}\right) \tag{114}$$

queries to $U_G^\top$.

The rest of the theorem is just to assemble all these steps. Note the requirement is

$$\|G - \nabla f(K)\|_F \leq \theta\|f(K)\|_F, \tag{115}$$

and according to Lemma 26, $\|\nabla f(K)\|_F \geq c\epsilon$ before convergence.

Denote the entries retrieved in the Entry Retrieving step form a matrix $\mathcal{G}$ where $\|\mathcal{G}\|_F = 1$, the norm estimated in the Norm Estimation step as $a_{\text{est}}$, our estimator is then $a_{\text{est}}\mathcal{G}$. Choose

$$\varepsilon_b = \frac{c\theta\varepsilon}{3}, \quad \varepsilon_a = \frac{c\theta\varepsilon}{3}, \quad \varepsilon_r = \frac{1}{3 + c\theta}, \tag{116}$$

we then know that $\|G - \nabla f(K)\| \leq \frac{c\theta\varepsilon}{3}$, $\nabla f(K) \geq c\varepsilon$, then

$$\mathcal{O}(\varepsilon) \leq \left|\|\nabla f(K)\|_F - \frac{c\theta\varepsilon}{3}\right| \leq \|G\|_F \leq \|\nabla f(K)\|_F + \frac{c\theta\varepsilon}{3}. \tag{117}$$

Thus it is clear to see

$$
\begin{aligned}
&\|a_{\text{est}}\mathcal{G} - \nabla f(K)\|_F \\
&= \left\|a_{\text{est}}\mathcal{G} - \|G\|_F\mathcal{G} + \|G\|_F\mathcal{G} - G + G - \nabla f(K)\right\|_F \\
&\leq \left\|a_{\text{est}}\mathcal{G} - \|G\|_F\mathcal{G}\right\|_F + \left\|\|G\|_F\mathcal{G} - G\right\|_F + \left\|G - \nabla f(K)\right\|_F \\
&\leq \varepsilon_a + \|G\|_F\varepsilon_r + \varepsilon_b \leq \varepsilon_a + (\|\nabla f(K)\|_F + \frac{c\theta\varepsilon}{3})\varepsilon_r + \varepsilon_b \\
&\leq \theta\|f(K)\|_F.
\end{aligned}
\tag{118}
$$

And the total gate complexity should be

$$\widetilde{\mathcal{O}}\left(\frac{a^{12}\rho^7\kappa^{11.5}m^{1.5}n}{\theta^{1.5}\varepsilon^{1.5}}\right). \tag{119}$$

$\square$

## H   Convergence analysis

**Lemma 32.** *Given any $K \in \mathcal{S}_K$, let $G$ be a $\theta$-robust estimate of $\nabla f(K)$. Then, we have*

$$
\begin{aligned}
\langle G, \nabla f(K)\rangle &\geq (1 - \theta)\|\nabla f(K)\|_F^2, \\
\|G\|_F^2 &\leq (1 + \theta)^2\|\nabla f(K)\|_F^2.
\end{aligned}
\tag{120}
$$

*Proof.* It is straightforward to verify that

$$\langle G, \nabla f(K) \rangle = \langle G - \nabla f(K) + \nabla f(K), \nabla f(K) \rangle = \langle G - \nabla f(K), \nabla f(K) \rangle + \|\nabla f(K)\|_F^2$$
$$\geq -\|G - \nabla f(K)\|_F \|\nabla f(K)\|_F + \|\nabla f(K)\|_F^2 \geq (1 - \theta)\|\nabla f(K)\|_F^2. \tag{121}$$

$$\|G\|_F^2 = \|G - \nabla f(K) + \nabla f(K)\|_F^2 \leq (\|G - \nabla f(K)\|_F + \|\nabla f(K)\|_F)^2$$
$$\leq (1 + \theta)^2 \|\nabla f(K)\|_F^2. \tag{122}$$

$\square$

**Lemma 33** (Descent lemma)**.** *Given $K \in \mathcal{S}_K(a)$, and let $G$ be a $\theta$-robust estimate of $\nabla f(K)$. Then, there exists a positive $\sigma_m$ such that for all $\sigma \in [0, \sigma_m]$, we have*

$$f(K - \sigma G) - f(K^*) \leq (1 - \sigma/\mu)(f(K) - f(K^*)), \tag{123}$$

*where $\mu$, $\sigma_m$ only depends on $A, B, R, Q$, and $a$, and $\mu > \sigma_m$.*

*Proof.* This lemma is a direct consequence of Lemma 32 (with $\theta < 1/2$) and [45, Proposition 6]. $\square$

**Proposition 34.** *For any initial stabilizing feedback gain $K_0 \in \mathcal{S}_K$, we denote $a := f(K_0)$. Then, there exists constants $\mu > \sigma_m > 0$, depending only on $A, B, Q, R$, and $a$, such that for any fixed $\sigma \in [0, \sigma_m]$, the iterates of Algorithm 1 satisfy*

$$f(K_k) - f(K^*) \leq (1 - \sigma/\mu)^k (f(K_0) - f(K^*)), \tag{124a}$$
$$\|K_k - K^*\|^2 \leq b(1 - \sigma/\mu)^k \|K_0 - K^*\|^2, \tag{124b}$$

*where $b$ is an absolute constant that is independent of $k$.*

*Proof.* The first part (124a) is a direct corollary of Lemma 33. By Lemma 9 and (124a), we have

$$\|K_k - K^*\|_F^2 \leq \frac{a}{\nu \lambda_{\min}(R)}(f(K_k) - f(K^*)) \tag{125}$$

$$\leq (1 - \sigma/\mu)^k (f(K_0) - f(K^*)) \leq b(1 - \sigma/\mu)^k \|K_0 - K^*\|, \tag{126}$$

where the last step follows from [45, Lamma 2] and $b = \lambda_{\max}(RX(K_0))$. $\square$

**Theorem 35** (Quantum policy gradient for LQR)**.** *Assume that we have efficient procedures (as described in Assumption 1) to access the problem data $A, B, Q, R$ in $\mathcal{O}(\text{poly}\log(n))$ time. Let $K_0 \in \mathcal{S}_K$ be a stabilizing policy. Then, Algorithm 1 outputs an $\varepsilon$-approximate solution to Problem 1 in cost*

$$\widetilde{\mathcal{O}}\left(\frac{m^{1.5}n}{\varepsilon^{1.5}}\right). \tag{127}$$

*Proof.* Notice that with lemma 33, we need $\log\left(\frac{1}{\varepsilon}\right)$ iterations. By Theorem 31, the quantum gradient estimation subroutine requires $\widetilde{\mathcal{O}}\left(\frac{m^{1.5}n}{\varepsilon^{1.5}}\right)$ elementary gates. $\square$

# I   Extended numerical experiments

## I.1   Aircraft Control Problem

We perform another experiment on a practical problem that can be formulated as LQR. Here, we consider the aircraft flight control problem, specifically for pitch angle control. We adopt a linearized model of the aircraft around a steady flight condition. For a small aircraft, the pitch dynamics can be represented by the following state variables: pitch angle $\theta$ (rad) and pitch rate $q$ (rad/s). The control input is elevator deflection angle $\delta$ (rad). The state-space model can be represented as $\dot{x} = Ax + Bu$, where $x = [\theta, q]^\top$, $u = [\delta]$. We set $A = [[0, 1], [0, -0.5]]^\top$, $B = [0, 1]^\top$, $Q = [[10, 0], [0, 1]]^\top$, and $R = [0.1]$. The plot of our optimization curve is available in the Figure Figure 3. Our method converges faster than the classical method.

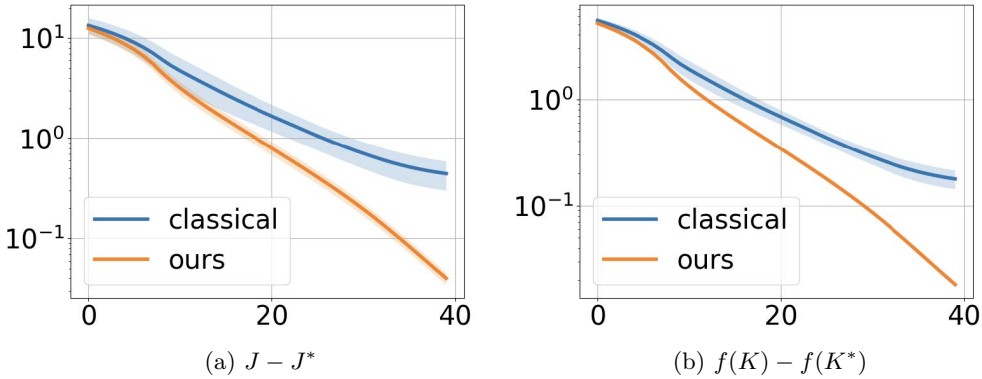

(a) $J - J^*$             (b) $f(K) - f(K^*)$

Figure 3: **Numerical Results on Convergence.** In the aircraft control problem, our policy gradient descent algorithm converges much faster than classic method [45].

## I.2 Relative Errors

We conducted further numerical experiments to understand how the optimality scales with problem size in Figure Figure 4. Here, we scale the number of masses in a spring-mass system from 2 to 4, and the problem dimension scales accordingly from 2 to 8. We measure the optimality by the relative error found in both our method and the classical method. The relative errors are $(J - J^*)/j^*$ and $(f(K) - f(K^*))/f(K^*)$. As the dimension scales up, the relative errors increase, while our method consistently outperforms the classical optimization method.

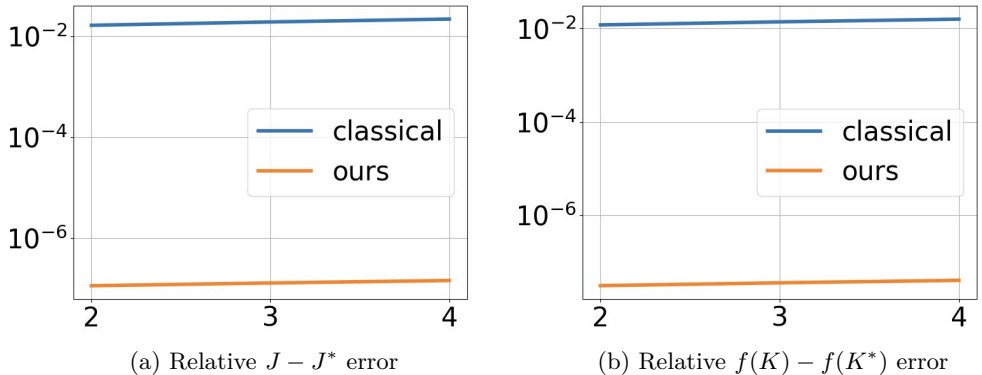

(a) Relative $J - J^*$ error      (b) Relative $f(K) - f(K^*)$ error

Figure 4: **Relative Error.** We scale the size of a mass-spring system and our method consistently gets smaller relative error compared to [45].

