# OpenReview forum: "Differentiable Quantum Computing for Large-scale Linear Control"
_NeurIPS.cc/2024/Conference — NeurIPS 2024 poster_

### Official Review · Reviewer_7tHw · 2024-07-11

**Soundness:** 3
**Presentation:** 3
**Contribution:** 3
**Rating:** 6
**Confidence:** 2

**Summary:**

This paper introduces an end-to-end quantum algorithm for linear quadratic control problem. The proposed quantum-assisted differentiable simulator is suitable for large-scale dynamical systems where the dimension of system state is huge. Sample complexity is also provided when apply quantum computation in such problem. Simulated results support its theory.

**Strengths:**

The quantum application to linear-quadratic problem seems new and provides a lot computation benefit.

**Weaknesses:**

This work only considers linear-quadratic control problem.

**Questions:**

What are the optimality plot such as the plots shown in Figure 2(a) and 2(b) for Figure 2(c) when you increase the system dimension?

---

> ### Author Rebuttal · Authors · 2024-08-05
>
> We thank the reviewer for recognizing the novelty and computational advantage of the proposed quantum application.
>
> Due to the page limit, we only consider the classic LQR problem in this paper. Nevertheless, our approach can be readily generalized to other optimal control problems, such as distributed LQR and nonlinear control problems. We plan to investigate these applications in future work.
>
> We conducted further numerical experiments to understand how the optimality scales with problem size, see Figure 2 in the uploaded PDF file. Here, we scale the number of masses $g$ from 2 to 4, and the problem dimension scales accordingly from 2 to 8. We measure the optimality by the relative error found in both our method and the classical method. The relative $f(K)$ error is $|f(K) - f(K^*)| / f(K^*)$ and relative $J$ error is $|J - J^*| / J^*$.
>
> The results are reported in the following table. As the dimension scales up, the relative errors increase, while our method consistently outperforms the classical optimization method. The plot has been added to the uploaded PDF file (see Figure 2).
>
>
> | problem dimension      | 2g = 2          | 2g = 6          | 2g = 8          |
> |---------------|------------------|------------------|------------------|
> | relative fK classical | 0.01177443       | 0.01369893       | 0.01562343       |
> | relative fK quantum   | 3.08486486e-08   | 3.54893993e-08   | 4.01301499e-08   |
> | relative J classical  | 0.01636588       | 0.01905698       | 0.02174809       |
> | relative J quantum    | 1.13132601e-07   | 1.28453806e-07   | 1.43775011e-07   |
>
> We sincerely thank the reviewer for the detailed review. We plan to address the reviewer’s questions in the camera-ready version by elaborating on the possible generalization of the proposed methodology and including the new numerical experiments. If the reviewer finds this additional information helpful, we kindly request consideration for an increased preliminary rating for this submission.

---

### Official Review · Reviewer_dSDS · 2024-07-13

**Soundness:** 2
**Presentation:** 2
**Contribution:** 2
**Rating:** 5
**Confidence:** 2

**Summary:**

The paper "Differentiable Quantum Computing for Large-scale Linear Control" introduces a quantum algorithm for linear-quadratic control problems, offering provable speedups. It utilizes a policy gradient method enhanced with a novel quantum subroutine for solving the matrix Lyapunov equation, leading to more accurate and robust gradient estimation than classical methods. The proposed algorithm achieves a super-quadratic speedup, making it the first end-to-end quantum application to linear control problems with demonstrable quantum advantages.

**Strengths:**

Quantum Speedup: The algorithm achieves a super-quadratic speedup over classical methods, which is a significant advancement in the field of quantum computing for control problems.

Innovative Approach: It introduces a novel quantum-assisted differentiable simulator, enhancing the accuracy and robustness of gradient estimation.

**Weaknesses:**

The experiments are insufficient. The proposed method has only been applied to simple abstract problems, but it would be beneficial to conduct experiments on practical problems in the form of LQR.

**Questions:**

1. How does the proposed quantum algorithm handle cases where the sparsity assumptions on matrices A,B,Q,R do not hold?

2. Can the authors provide detailed runtime performance comparisons between the proposed quantum algorithm and state-of-the-art classical algorithms?

3. Comparison with Other Methods: How does the proposed method's stability and convergence rate compare with other existing quantum and classical approaches for solving linear-quadratic control problems?

**Limitations:**

The primary limitation of the proposed method is its dependency on the availability of quantum resources and the sparsity assumptions for the matrices involved.

---

> ### Author Rebuttal · Authors · 2024-08-05
>
> We appreciate the reviewer recognizing the super-quadratic speedup achieved in this paper as a significant advancement in quantum computing for control problems.
>
> Regarding the reviewer’s comment on the insufficient experiments as a main weakness of this paper, while the key contribution of this work is on the analytical formulation and theoretical advances, we conducted several experiments in the original submission.
>  We perform another experiment on a practical problem that can be formulated as LQR, see Figure 1 in the uploaded PDF. Here, we consider the aircraft flight control problem, specifically for pitch angle control. We adopt a linearized model of the aircraft around a steady flight condition. For a small aircraft, the pitch dynamics can be represented by the following state variables: pitch angle $\theta$ (rad) and pitch rate $q$ (rad/s). The control input is elevator deflection angle $\delta$ (rad). The state-space model can be represented as
> $\dot{x} =Ax+Bu$, where $x = [\theta, q]^T$, $u = [\delta]$.
> We set $A=[[0, 1], [0, -0.5]]^T$, $B=[0, 1]^T$, $Q=[[10, 0], [0, 1]]^T$, and $R=[0.1]$.
> The plot of our optimization curve is available in the uploaded PDF (see Figure 1). Clearly, our method *converges faster than the classical method*.
>
>
> Regarding the question on the sparsity assumption, *the sparsity assumptions are standard* across quantum algorithms research and are *necessary to efficiently load classical data into quantum registers*. Since our primary objective of this work is to establish the theoretical advantage of the proposed quantum algorithm, the sparsity assumptions are reasonable and at par with other works in this field. In practice, even when the sparsity assumptions do not hold, it is still possible to extend our algorithmic design to other types of quantum input models that allow dense data (e.g., [1, 2]). We will add this information to justify our sparse matrix input model and leave the adaptation of other input models in future work.
>
>
> Since the proposed quantum algorithm utilizes advanced subroutines such as linear combination of unitaries (LCU), a detailed runtime analysis would require developing a customized compiler to estimate the gate count, which is clearly beyond the scope of the current paper. The **asymptotic analysis in the paper asserts a super-quadratic speedup against SOTA classical** algorithms, which paves the way toward a comprehensive resource analysis in future research.
>
>
> Regarding the comparison with other methods: to our best knowledge, this paper is the **first to prove the robust convergence to the LQR solution in the literature of quantum computation**. We achieve a linear convergence rate, which is comparable with the classical SOTA, while our quantum advantage comes from the fact that we can **solve the matrix Lyapunov equation exponentially faster than any classical means**. Since our algorithm uses analytical gradient estimation, it **demonstrates stability in the SGD iterations** and achieves **faster convergence compared to classical model-free methods**, as illustrated in our numerical experiments (see figure 2 in the main paper).
>
> We sincerely thank the reviewer for the detailed review. We plan to address the reviewer’s questions in the camera-ready version by elaborating on the technical assumptions and adding a numerical experiment with a practical background. If the reviewer finds this additional information helpful, we kindly request consideration for an increased preliminary rating for this submission.
>
> [1] Wang and Wossnig (2018). A quantum algorithm for simulating non-sparse Hamiltonians. [arXiv:1803.08273](https://arxiv.org/abs/1803.08273)
>
> [2] Liu and Lin (2023). Dense outputs from quantum simulations. [arXiv:2307.14441](https://arxiv.org/abs/2307.14441)

---

> > ### Comment · Reviewer_dSDS · 2024-08-13
> > **Thank you for the responses.**
> >
> > The author's response has somewhat addressed my concerns and questions. I will increase my score accordingly.

---

> > > ### Author Response · Authors · 2024-08-14
> > > **Thanks for your kind response.**
> > >
> > > Thank you for your thoughtful feedback and for taking the time to review our work. We sincrely appreciate your willingness to reconsider the score!

---

### Official Review · Reviewer_v7Mi · 2024-07-13

**Soundness:** 3
**Presentation:** 3
**Contribution:** 3
**Rating:** 5
**Confidence:** 3

**Summary:**

This proposes an end-to-end solution to the quantum-assisted LQR problem. Based on a policy gradient method, the proposed algorithm incorporates a quantum subroutine for solving the matrix Lyapunov equation, achieving a super-quadratic speedup.

**Strengths:**

1. To the best of my knowledge, this is the first end-to-end quantum application to linear control problems with provable quantum advantage.

2. This paper also provides numerical evidence to demonstrate the robustness and favorable convergence behavior of the method.

3. This paper is clearly written.

**Weaknesses:**

1. While the paper highlights the theoretical advantages of the quantum algorithm, implementing these algorithms on current quantum hardware might pose significant challenges. Today's quantum computers suffer from noise and have a limited number of qubits, which could affect the actual performance and reliability of the algorithm.

2. Although the paper claims a super-quadratic speedup, it's important to verify whether this speedup is practically achievable. Especially in large-scale industrial models, the real-world complexity and scalability of the algorithm are critical issues.

3. This paper is not always well written. For example, in Line 520, "The evolution of a quantum state can always described by a unitary operator".

**Questions:**

1. How does the proposed quantum algorithm account for the current limitations of quantum hardware, such as noise and the limited number of qubits?
2. The paper introduces a novel quantum subroutine for solving the matrix Lyapunov equation. Could you elaborate on the specific technical innovations that this subroutine brings compared to existing quantum algorithms? What are the key theoretical breakthroughs that enable the claimed super-quadratic speedup?

**Limitations:**

The limitations of this work have been adequately discussed.

---

> ### Author Rebuttal · Authors · 2024-08-05
>
> We appreciate the reviewer recognizing our work as the first end-to-end quantum application to linear control problems with a provable quantum advantage.
>
> One main concern in the review is related to the performance and reliability of the proposed quantum algorithm on noisy quantum hardware. The primary objective of this work is to establish the theoretical advantage of our algorithm, and the core subroutine (i.e., solving the matrix Lyapunov equation) requires a fault-tolerant quantum computer. That being said, our method still has reasonable potential for implementation on early fault-tolerant devices. In particular, the following aspects of our algorithm make it particularly robust given **mild noise** and a **limited number of logical qubits**:
>
> 1. The **hybrid quantum-classical nature** of our algorithm *mitigates noise* by limiting the runtime of each quantum processing routine via interspersed classical routines, which is advantageous on quantum devices with fewer qubits and shorter qubit coherence times.
> 2. The classical optimizer in our algorithm utilizes a **stochastic gradient descent (SGD)** algorithm, which *in principle converges for any unbiased gradient estimator*. The unbiasedness is a reasonable assumption given the independent nature of quantum noise. Therefore, it is expected that our algorithm is resistant to (sufficiently low amounts of) independent noise in gate execution and/or memory.
> 3. Our algorithm is truly **end-to-end** in the sense that the input and output are both classical data, which is often not the case for other proposed quantum algorithms for optimal control and RL. For example, most cited works in the “Quantum reinforcement learning” part under Section 2 do not support a comparable classical output.
>
>
> The second question concerns the key theoretical innovation in the novel quantum subroutine for solving the matrix Lyapunov equation. This quantum subroutine is **exponentially faster** than its classical counterpart and is the key ingredient in achieving *super-quadratic quantum speedup*. The high-level idea of the quantum subroutine is to use the linear combination of unitaries (LCU) technique to compute the integral formula (12). To do so, we need an efficient implementation of the block-encoded operator $\exp(\mathcal{A}t)$, where $\mathcal{A}$ is Hurwitz. Note that no explicit upper bound on the largest singular value of $\mathcal{A}$ is known. Most existing quantum algorithms for this task (i.e., to compute $\exp(\mathcal{A}t)$) cannot be applied in our case, either because they have stronger assumptions on the matrix $\mathcal{A}$ (Quantum Singular Value Transformation (QSVT) [1] and Linear Combination of Hamiltonian Simulations (LCHS) [2]) or because they have worse asymptotic scaling (Taylor series expansion leads to an exponentially small success rate, and the so-called time-marching strategy [3] has super-linear scaling in $t$ (more precisely, $t^2$). Instead, we employ some ideas from Quantum EigenValue Transformation (QEVT) [4], a recent breakthrough in quantum algorithms, to construct this block-encoded operator with favorable asymptotic scaling (linear in $t$ and polylogarithmic in $1/\epsilon$). It is worth noting that our construction is not identical to the one in [4], as they do not have an explicit block-encoding form in the original paper.
>
> We sincerely thank the reviewer for the detailed review. We plan to address the reviewer’s questions in the camera-ready version of this paper by further elaborating on our technical novelty and near-term feasibility. If the reviewer finds this additional information helpful, we kindly request consideration for an increased preliminary rating for this submission.
>
> [1] Gilyén et al. (2018). Quantum singular value transformation and beyond: exponential improvements for quantum matrix arithmetic. [arXiv:1806.01838](https://arxiv.org/abs/1806.01838)
>
> [2] An et al. (2023). Linear combination of Hamiltonian simulation for nonunitary dynamics with optimal state preparation cost. [arXiv:2303.01029](https://arxiv.org/abs/2303.01029)
>
> [3] Fang et al. (2022). Time-marching based quantum solvers for time-dependent linear differential equations. [arXiv:2208.06941](https://arxiv.org/abs/2208.06941)
>
> [4] Low and Su. (2024). Quantum eigenvalue processing. [arXiv:2401.06240](https://arxiv.org/abs/2401.06240)

---

> > ### Comment · Reviewer_v7Mi · 2024-08-11
> >
> > Thank you for your detailed rebuttal and for addressing the concerns raised. I appreciate the insights provided, particularly regarding the theoretical contributions and potential robustness of your proposed quantum algorithm.
> >
> > Your work establishes a good theoretical foundation, and the practical realization and scalability of these methods in real-world scenarios are important aspects to consider moving forward. I will maintain my current assessment of the submission.

---

> > > ### Author Response · Authors · 2024-08-13
> > > **Thanks for your comment!**
> > >
> > > Thank you for your thoughtful review and for recognizing the theoretical contributions of our work. We appreciate your insights on the practical realization and scalability of our proposed methods. Your feedback has been important in refining our approach.

---

### Official Review · Reviewer_oizh · 2024-07-16

**Soundness:** 2
**Presentation:** 3
**Contribution:** 3
**Rating:** 5
**Confidence:** 2

**Summary:**

This paper studies the problem of applying quantum computing to linear quadratic regulator (LQR) control. The approach is based on an efficient quantum estimation of the policy gradient. When the dimension n of the state space is large, the proposed approach can achieve orders of magnitude improvement on the time complexity to find the optimal controller compared to existing policy gradient methods.

**Strengths:**

I am not familiar with quantum computing. Since the LQR sample complexity has received much attention recently, this work should be interesting for the learning theory community if the claim about time complexity improvement is correct.

**Weaknesses:**

Since this work proposes a model-based approach and assumes access to the exact model, including A, B, Q, and R (Algorithm 1), I wonder why the authors compare with the model-free approach in [44], which does not assume knowledge of the model and uses two-point gradient estimation. I don’t think this is a fair comparison.

Besides, I do not understand why the application of the proposed quantum approach is limited to the classic LQR problem. If the real contribution is the first quantum approach to solve the Lyapunov equation (11) efficiently, I think the proposed method can be applied to other important problems like stability analysis. I hope the authors can clarify the most general statement of the main contribution or what limits the application to LQR.

**Questions:**

Besides my comments in the weakness part, I also hope the authors can clarify if the proposed method is robust to estimation errors in A and B if we estimate them from samples (see [Dean, Sarah, et al., 2020]).

[Dean, Sarah, et al., 2020] Dean, Sarah, et al. "On the sample complexity of the linear quadratic regulator." Foundations of Computational Mathematics 20.4 (2020): 633-679.

**Limitations:**

I do not see any potential negative societal impact of this work.

---

> ### Author Rebuttal · Authors · 2024-08-05
>
> We thank the reviewer for confirming that our work is of interest to the learning theory community, given that the technical part is sound. In what follows, we address the concerns regarding the weaknesses and technical details mentioned in the review.
>
> In Table 1, the “(Model-based) policy gradient [44]” item summarizes the convergence results proven in [Theorem 1, 44]. This result is proven under the assumption of a “known model,” indicating that the policy gradient can be computed explicitly. It focuses on the convergence rate of an exact gradient descent method and is parallel to the “model-free” approach discussed later in the same paper.  Given context in [44], since our algorithm adopts a similar policy gradient idea with quantum-assisted gradient estimation, we believe that it is a fair comparison to mention the (model-based) policy gradient result in Table 1. We apologize for the confusion and will provide expanded elaboration in the camera-ready version.
>
> We agree with the reviewer that our quantum subroutine for the matrix Lyapunov equation is of *independent interest* and its **application need NOT be limited to the LQR problem**. Our quantum algorithm is based on the integral formula (12),
>
> $$X^* = \int^\infty_0 e^{\mathcal{A}t} \Omega e^{\mathcal{A}^T t},$$
>
> which holds only when the matrix $\mathcal{A}$ is Hurwitz stable. If our quantum algorithm does not produce a correct solution to the Lyapunov equation in a designated time, we may conclude that the matrix $\mathcal{A}$ is non-stable. However, the output of our quantum algorithm is a “block-encoded matrix,” a quantum circuit that cannot be efficiently simulated by classical means. This means that checking if the output “solution” satisfies the Lyapunov equation is quite non-trivial and requires additional algorithmic design, as opposed to the classical case where we can do simple matrix multiplication. For the sake of conciseness and self-consistency of this paper, we mainly focus on the end-to-end solution of the LQR problem. We will clarify our main technical contribution by discussing the potential applications of our quantum subroutine in the camera-ready version and leave the technical details for future work.
>
> Regarding the estimation error on $A$ and $B$: if the matrix $A$ and $B$ are estimated using the independent data collection scheme (as indicated in [Dean, Sarah, et al., 2020]) and these estimates allow efficient quantum input procedures (as described in Assumption 1 in our manuscript), we believe that our method is robust to the estimation error. This is because the relative error in the LQR objective function can be characterized similarly to how it is presented in Proposition 1.2 in the reference paper [Dean, Sarah, et al., 2020]. We thank the reviewer for making this insightful and interesting observation.  We will cite this paper in the camera-ready version.
>
> We sincerely thank the reviewer for the detailed review. We plan to address the reviewer’s questions in the camera-ready version by clarifying the comparison in Table 1 and elaborating on the scope of potential applications. If the reviewer finds this additional information helpful, we would be grateful if they could consider an increased preliminary rating for this submission.

---

### Author Rebuttal · Authors · 2024-08-05

We sincerely thank all the reviewers for their invaluable comments on our submission. We are particularly grateful for the reviewers' recognition of the novelty of our paper as the *first end-to-end quantum application to optimal control* (Reviewers v7Mi, 7tHw), the acknowledgment of the *super-quadratic speedup* over classical methods as a significant advancement (Reviewer dSDS), and the overall *interest our work generates within the broader learning theory community* (Reviewer oizh).

In this message, we would like to address a few questions raised by the reviewers regarding the technical contributions, algorithm applicability, and numerical experiments.

The proposed algorithm is truly **end-to-end**, with classical input and classical output, and achieves a **super-quadratic speedup** against SOTA classical methods for the LQR problem. Both aspects make it a significant result in quantum computing. Most existing quantum algorithms for classical optimization and control problems either return a quantum state as output, requiring an exponential overhead to convert to classically readable data, or only achieve a quadratic speedup by leveraging a Grover-type quantum algorithm. In contrast, our super-quadratic speedup comes from an exponentially faster quantum subroutine for solving the matrix Lyapunov equation. To the best of our knowledge, this matrix Lyapunov equation solver is a novel quantum algorithm and is of independent interest. As Reviewer oizh mentioned, this subroutine can be applied to other problems, such as stability analysis. The key technical difficulty behind this new quantum subroutine is that it requires efficiently computing the matrix exponential of a Hurwitz matrix, for which most existing quantum algorithms are not directly applicable. We employ ideas from a recent result known as Quantum EigenValue Transformation (QEVT) to resolve this difficulty.

Since the primary goal of this paper is to investigate the theoretical advantage of the proposed quantum algorithm, we adopt the sparse-matrix input model, which is considered one of **the most standard input models for quantum algorithm design**. The hybrid quantum-classical nature of our algorithm limits the runtime of each quantum processing routine by interspersing classical routines, which is advantageous on near- and mid-term quantum devices with fewer qubits and shorter qubit coherence times. While estimating the actual gate count is beyond the scope of this paper, the hybrid quantum-classical approach makes our algorithm more feasible than many other algorithms without a classical component.
Additionally, we add two more numerical experiments to show the scalability and practical relevance of our approach. We observe that (1) our method **converges faster** than classical methods in an **aircraft flight control** problem, and (2) our method **consistently outperforms classical methods** as the **problem dimension increases**. The figures are included in the uploaded PDF file. We will add these results in the camera-ready version of this submission.

We appreciate the reviewers' informative feedback and look forward to further discussions on specific technical or conceptual questions. If this additional information proves helpful, we would be thankful if the reviewers could consider a higher preliminary rating for our submission.

---

### Decision · Program_Chairs · 2024-09-25

**Decision:**

Accept (poster)

**Comment:**

The paper considers the problem of finding an optimal controller for the continuous LQR control problem. They take a gradient descent approach to the problem which is known to have linear convergence from prior work. The main contribution of the paper is to provide a quantum algorithm for the estimation of the gradient which can run in time linear in the matrix dimension (as opposed to a classical algorithm which requires a cube of the dimension).

Overall the paper provides a first foray of quantum algorithms into the fundamental problem of LQR control showing a quadratic speedup under their assumptions. While I do not have enough expertise to evaluate the quantum assumptions, the reviewers all found the paper to be reasonable. I recommend accept.